# Ovule Transcriptome Analysis Discloses Deregulation of Genes and Pathways in Sexual and Apomictic *Limonium* Species (Plumbaginaceae)

**DOI:** 10.3390/genes14040901

**Published:** 2023-04-12

**Authors:** Ana D. Caperta, Isabel Fernandes, Sofia I. R. Conceição, Isabel Marques, Ana S. Róis, Octávio S. Paulo

**Affiliations:** 1Linking Landscape, Environment, Agriculture and Food (LEAF), Research Center, Associate Laboratory TERRA, Instituto Superior de Agronomia (ISA), Universidade de Lisboa, Tapada da Ajuda, 1349-017 Lisboa, Portugalrois.ana@gmail.com (A.S.R.); 2cE3c—Centre for Ecology, Evolution and Environmental Changes, CHANGE—Global Change and Sustainability Institute, Faculdade de Ciências, Universidade de Lisboa, 1749-016 Lisboa, Portugal; isabelcmoniz@gmail.com; 3LASIGE Computer Science and Engineering Research Centre, Faculdade de Ciências, Universidade de Lisboa, 1749-016 Lisboa, Portugal; 4Forest Research Centre (CEF), Associate Laboratory TERRA, Instituto Superior de Agronomia (ISA), Universidade de Lisboa, Tapada da Ajuda, 1349-017 Lisboa, Portugal; 5School of Psychology and Life Sciences, Universidade Lusófona de Humanidades e Tecnologias (ULHT), Campo Grande 376, 1749-024 Lisboa, Portugal

**Keywords:** apomixis, floral development, functional annotation, Illumina sequencing, MADS-box, male sterility, sea lavenders, sexual reproduction, transcription factors

## Abstract

The genus *Limonium* Mill. (sea lavenders) includes species with sexual and apomixis reproductive strategies, although the genes involved in these processes are unknown. To explore the mechanisms beyond these reproduction modes, transcriptome profiling of sexual, male sterile, and facultative apomictic species was carried out using ovules from different developmental stages. In total, 15,166 unigenes were found to be differentially expressed with apomictic vs. sexual reproduction, of which 4275 were uniquely annotated using an *Arabidopsis thaliana* database, with different regulations according to each stage and/or species compared. Gene ontology (GO) enrichment analysis indicated that genes related to tubulin, actin, the ubiquitin degradation process, reactive oxygen species scavenging, hormone signaling such as the ethylene signaling pathway and gibberellic acid-dependent signal, and transcription factors were found among differentially expressed genes (DEGs) between apomictic and sexual plants. We found that 24% of uniquely annotated DEGs were likely to be implicated in flower development, male sterility, pollen formation, pollen-stigma interactions, and pollen tube formation. The present study identifies candidate genes that are highly associated with distinct reproductive modes and sheds light on the molecular mechanisms of apomixis expression in *Limonium* sp.

## 1. Introduction

Apomixis (agamospermy), the asexual seed production found in less than 1% of angiosperms [1], can be either independent of or dependent on pollination [2,3]. Most natural apomicts produce meiotic-reduced pollen involved in the fertilization of the polar nuclei in the embryo sac (pseudogamy) [2,3], but others reproduce independently of pollination for the initiation of the embryo or endosperm formation (autonomous apomicts) [4,5].

Different from sexual reproduction, apomixis is characterized by alterations in the developmental program during the formation and development of the female germline [2,6]. Apomicts can reproduce via gametophytic apomixis, involving the formation of an unreduced embryo sac (apomeiosis) that gives rise to a parthenogenetic embryo and a functional endosperm without the 2 maternal: 1 parental genome ratio [2,6]. The unreduced gametophytes can develop via restitutional meiosis or mitotic division (diplospory) or by a somatic, unreduced cell of the nucellus, which develops into an embryonic sac (apospory) [2,6]. In sporophytic apomicts (adventitious embryonies), the embryos originate from somatic tissues of the nucellus and/or integument cells. The formation of these apomictic embryos usually occurs in parallel with the formation of sexual embryos [7]. 

In gametophytic apomicts, pseudogamy is prevalent among aposporous apomicts, whereas autonomous endosperm formation seems to be prevalent in diplosporous apomicts [2,8]. These latter apomicts seem to be tolerant of deviations from a 2 maternal: 1 paternal genome contribution in the endosperm [4,5]. Autonomous apomicts tend to produce pollen with low viability and can even be male-sterile [9,10,11,12,13]. These apomicts include species from Asteraceae (e.g., *Crepis*, *Taraxacum* [9,10,11]), Plumbaginaceae (*Limonium* [12,13], Melastomataceae (*Miconia* [14], Poaceae (*Calamagrostis* [15]), and Rosaceae (*Alchemilla* [16]) genera, among others.

The emergence of apomixis in natural systems has been a long-standing topic of debate. It was hypothesized that the different types of apomixis are caused by different mutations that destabilize meiosis (megasporogenesis), the gametophyte (embryo sac), and egg formation [2,6]. Loci genetically linked to components of apomixis have been identified in various species, and sequencing of these loci has revealed several genes with the potential to play critical roles in apomixis [17,18]. It was hypothesized that apomixis could be caused by asynchronously expressed germline genes in the ovules of certain hybrids [19]. Transcriptome comparisons show that the genetic control of apomixis in gametophytic apomicts has been related to a wide range of mechanisms regulating gene expression, including protein degradation, transcription, cell cycle control, stress response, hormonal pathways, cell-to-cell signaling, and epigenetic mechanisms [18,20,21]. Several common genes found to be differentially expressed in multiple stages of apomictic and sexual seed production support the view that sexual and apomictic reproduction are closely related developmental pathways [22]. Recent studies present substantial evidence in support of a polyphenic condition for meiosis and determine that polyphenic shifts from apomeiosis to meiosis and vice versa are regulated by metabolic states [23].

The genus *Limonium* is a remarkable case study that could help with the identification of genetic-molecular factors potentially underlying apomixis [24]. The genus comprises c. 350 species with sexual and apomixis reproductive modes [25], having triploid and tetraploid apomicts with very large distributions in the Mediterranean region and the Atlantic coast [26,27,28]. The mode of speciation in the genus has been hypothesized to combine a polymorphic sexual system, hybridization under alloploid conditions, polyploidy combining autoploidy (unreduced pollen), allopolyploidy, and apomixis [13,24,25,29,30,31]. The polymorphic sexual system is associated with flower polymorphisms (ancillary pollen and stigma and/or heterostyly) and self-incompatibility (SI) under sporophytic control [29,30]. Coarsely reticulate pollen grains germinate on papillose stigmas and finely reticulate pollen grains germinate on cob-like stigmas, while the reverse combinations result in unsuccessful fertilization. Dimorphic SI species have two pollen stigma combinations and reproduce sexually as in diploids (*Limonium ovalifolium*, 2*n* = 16 chromosomes). Monomorphic self-compatible species present self-fertile combinations, while monomorphic SI species show only one pollen-stigma combination and produce seeds through apomixis as in tetraploids (*Limonium multiflorum*; *Limonium dodartii*, 2*n* = 35, 36) [12,28]. Sexual species form meiotically reduced tetrasporic embryo sacs [32,33,34] as in *L. ovalifolium* [13]. Whereas triploid and tetraploid facultative apomicts originate both reduced and unreduced, diplosporic apomictic embryo sacs (*Limonium oleifolium* (syn. *Statice oleaefolia*) [33]; *Limonium transwallianum* [34]. Pollen in *Limonium* apomitics ranges from low to high fertility or is not produced at all [12,25,31,35]. In the agamospermous species of the *L. binervosum* group (e.g., *Limonium binervosum s.s*., *L. dodartii*, and *L. multiflorum*), male sterility, i.e., lack of viable pollen, is widespread, and male sterile colonies are confined to defined taxa/areas [12,13,26,28,36]. Male-sterile *L. multiflorum* plants from diverse colonies present aborted pollen with collapsed morphology without the typical exine patterns, pointing to a sporophytic defect [12]. The elevated number of seeds with high viability formed by this species seem to be the result of autonomous apomixis [13].

In this study, we aim to characterize the genetic factors implicated in *Limonium* reproductive modes by identifying the genes that are differentially expressed in ovules during sexual and apomixis seed production. Ovules in different stages of development were extracted to compare sexual (*Limonium auriculifolium* (syn. *Limonium nydeggeri* [37]), *L. ovalifolium*), putative facultative apomictic (*L. dodartii*), and male-sterile (*L. multiflorum*) plants, previously characterized cytogenetically, cytometrically, genetically, and reproductively [12,13,24,31] and used for profiling through RNA sequencing (RNA-Seq). This technology allows the identification of DEGs and the inference of their expression with high accuracy [38]. Downstream analysis, including functional annotation of the assembled transcriptome and GO enrichment analysis of the annotated DEGs, was used to provide meaningful biological insights that contribute to a better understanding of the molecular mechanisms and pathways that control the switch from sexual reproduction to apomixis in the organisms under study. Our experimental approach was designed to overcome difficulties due to the occurrence of differing patterns among reproductive modes. The specific goals of this study were to: (1) identify transcripts showing differential expression between male-sterile *L. multiflorum* and sexual plants; (2) partition DEGs into groups whose fold-changes reflect genes potentially involved in flower development; and (3) frame our findings according to previous and ongoing studies to understand apomixis regulation in *Limonium*.

## 2. Materials and Methods

### 2.1. Plant Material

Plants from four *Limonium* species were selected to represent different reproductive modes, i.e., diploid sexual as *L. auriculifolium* (*n* = 2) and *L. ovalifolium* (*n* = 2; 2*n* = 2x = 16 chromosomes) [12,13], tetraploid apomictic as *L. multiflorum* (*n* = 2; 2*n* = 4x = 35), and the facultative apomictic *L. dodartii* (*n* = 1; 2*n* = 4x = 35) [13,24]. These plants, grown from seedlings raised from seed collected in the wild, were maintained at the ex situ collection in a semi-closed greenhouse at the Instituto Superior de Agronomia, Lisbon (Table 1). 

### 2.2. Ovule Extraction

Flower buds at distinct developmental stages were sampled prior to anthesis according to their size (between 2 and 5 mm) based on cytoembryological observations as in [13], at standardized times (between 9:00 a.m. and 12:00 p.m.). The ovules were selected with respect to the timing of apomeiosis (Stage 1—S1), megagametogenesis (embryo sac, Stage 2—S2), parthenogenesis, and endosperm formation (Stages 3/4—S3/S4), detailed in [13]. Dissection of ovules was performed in a sterile laminar air flow cabinet under a stereoscopic microscope (Stemi 2000-C, Zeiss) with the aid of tweezers and pencil-point spinal needles (Transmed). In total, 280 ovules were extracted from each ovary, containing one ovule each, and about ten to twenty ovules per stage were isolated and placed in a sterile Petri dish with B5 medium [39] to maintain hydration before RNA extraction. This procedure generated a total of 18 samples, including nine samples of ovules from apomictic plants (four biological replicates in stage S1 and five biological replicates in stage S2) and three samples from each of the remaining species (sexual: two species x three stages—S1, S2, and S3/S4; facultative apomictic: three biological replicates in stage S3/S4) (Table 1).

### 2.3. Total RNA Extraction and Library Preparation

Total RNA was extracted from all samples using the Spectrum™ Plant Total RNA Kit (Sigma-Aldrich, St. Louis, MO, USA). Nevertheless, some modifications were required for obtaining high-quality RNA, as detailed. Samples were collected in 150 µL of lysis solution and grounded with the help of a micropestle in the microfuge tube. Then, another 300 μL of lysis solution supplemented with β-mercaptoethanol was added to the tube, which was vortexed vigorously and incubated at 56 °C for 5 min. The lysate was centrifuged at 14,000 rpm for 2 min, and then the supernatant was transferred into a filter column and centrifuged for 1 min at 14,000 rpm. The clarified flow-through lysate was transferred to a new tube, and 250 μL of Binding Solution was added. The mixture was applied to a binding column and centrifuged for 1 min at 14,000 rpm. The remaining steps followed the manufacturer’s instructions. The quantity of RNA was determined using a BioDrop cuvette (BioDrop, Cambridge, UK) and electrophoresis on a 1% agarose gel. The RNA integrity number (RIN) was determined using an Agilent 2100 Bioanalyzer (Agilent Technologies, Santa Clara, CA, USA) and ranged from 9.19 to 9.45. The messenger RNA (mRNA) libraries were constructed with the Illumina “TruSeq Stranded mRNA Sample Preparation kit” (Illumina, San Diego, CA, USA) and sequenced on an Illumina NovaSeq6000 2× 100 bp at Macrogen facilities (Macrogen, Geumcheon-gu, Seoul, Korea).

### 2.4. Processing, Mapping, and Quantification of Illumina Reads

All raw reads have been deposited in the NCBI Sequence Read Archive (SRA), BioProject accession PRJNA752506. Quality control of the raw reads, including contaminants survey, was performed using FastQC version 0.11.9 [40] and FastQ Screen version 0.14.0 [41], which ran against the genomes of their default pre-indexed species and adaptors. Then, since all raw reads presented a quality base score over 36, Trimmomatic version 0.39 [42] was used to eliminate adaptors and filter reads of length below 36 base pairs (bp). A de novo transcriptome assembly was performed using Trinity version 2.11.0 [43], in which cleaned reads from all samples were combined to generate one global assembly since this software has shown consistent performance and has a high read alignment rate [44]. The assembly was assessed for completeness using BUSCO version 5 [45] through gVolante2 [46]. After alignment against the transcriptome using Bowtie2 aligner version 2.3.5 [47], sequences were quantified at gene-level expression with RSEM version 1.3.3 [48] through the Trinity pipeline. A principal component analysis (PCA) was performed to survey the relatedness of normalized gene counts using the function plotPCA in R Studio version 4.0.2 [49].

### 2.5. Differentially Expressed Genes Detection

To study significant differences between apomictic and sexual plants, differential expression analysis was performed with edgeR version 3.30.3 [50], which is a flexible empirical Bayes approach that uses weighted likelihood methods to estimate gene-specific variation even with very few or no replicates [51]. Overall, when studying differences between reproductive strategies, apomictic plants were set as the samples to test, while sexual and facultative apomictic plants were set as the controls, according to each comparison (Table 1). As such, up-regulated DEGs are more expressed in apomictic than in sexual plants, while down-regulated DEGs are less expressed in apomictic and more expressed in sexual plants.

Genes with a normalized |log2 fold change (log2 FC)| > 2 were defined as differentially expressed and used in the downstream analysis. In the comparison between apomictic and facultative apomictic plants, in which all samples have at least 3 replicates, DEGs were previously filtered by *p* < 0.01. Venn diagrams were used to plot DEGs between different comparisons through matplotlib version 3.3.3 [52] in Python version 3.9.0 (Python Software Foundation 2020). Additionally, DEGs commonly triggered by more than one comparison were searched for opposite regulation. 

### 2.6. Functional Annotation

Functional annotation of DEGs was performed with the Basic Local Alignment Search Tool (BLAST) version 2.10.1 command-line tool from the NCBI C++ Toolkit (National Center for Biotechnology Information 2020). Blastx was used to map DEGs to *A. thaliana* homologs against a local Swissprot database, filtering gene hits by a maximum E-value of 1.0E^−3^ and a minimum identity of 40%. [53]. Then, to avoid duplicated results, DEGs annotated to the same *A. thaliana* homolog were filtered by identities and sequence length, keeping the transcripts with the highest values.

Housekeeping genes (HKGs) are typically required for the maintenance of basal cellular functions that are essential for its existence, regardless of their specific role in the tissue or organism. Since these genes are usually highly conserved, genes stably transcribed in all comparisons were filtered. Additionally, DEGs were searched for the most common housekeeping homolog genes in *A. thaliana* to study their potential role in reproduction in *Limonium* plants.

In addition, DEGs were searched for sRNA biogenesis, which is known to play pivotal roles in reproductive development [54,55], and oxidative stress-related genes, which negatively affect reproductive development in plants [56]. DEGs associated with tryptophan and ethylene metabolism were investigated, which are associated with the biosynthesis and regulation of the phytohormone auxin, a vital component of plant reproduction since it regulates both male and female reproductive organs [57,58]. Moreover, DEGs related to aminoacyl-tRNA metabolism and lysine degradation, which are respectively essential to produce ribosomes and proteins, and epigenetic processes through DNA methylation [59,60] were searched.

### 2.7. Transcription Factors (TFs) Involved in Plant Reproduction

TFs can be engaged in plant reproduction, namely in flower development such as *APETALA* (AP genes: *AP1, AP2*, and *AP3*), *PISTILLATA* (*PI*), *SEPALLATA* (SEP genes: *SEP1*, *SEP3*), and other MADS-box TFs [61,62,63], and in male sterility (e.g., *ROS1*, *DMC1*, *MS2*, *POP1*, and *4CLL1*; [64,65]. As such, these genes, along with a list of *A. thaliana* TFs retrieved from the Plant Transcription Factor and Protein Kinase Identifier and Classifier database (iTAK v18.12) [66], were searched among DEGs to find if they were down-regulated in apomictic plants. Next, KEGG and WikiPathways enrichment analysis was performed with gProfiler to find relevant metabolic pathways among these TFs, following the same parameters mentioned for GO analysis (see below). Furthermore, uniquely annotated lists of DEGs were searched for GO terms related to pollen, such as the direct and child terms of the biological processes “microsporocyte differentiation” (GO:0010480), “pollen development” (GO:0009555), “pollen wall” (GO:0043667), “pollen coat” (GO:0070505), “pollination” (GO:0009856), and the “cellular components pollen tube” (GO:0090406), according to UniProtKB and StringDB.

### 2.8. Enrichment Analysis

Uniquely annotated DEGs were characterized with GO terms using the REST API on the UniprotKB website (The Uniprot Consortium, 2019). Finally, GO enrichment analyses were applied to log2FC-ordered lists of DEGs through an over-representation analysis (ORA) using the g:GOSt functional profiling tool from the gProfiler website [67], with the g:SCS tailored algorithm under FDR < 0.01, using a predefined *A. thaliana* custom background including only genes expressed by the samples in analysis. Enrichment results were summarized using REVIGO [68] through the removal of redundant GO terms with allowed similarity = 0.5 and then plotted with the R ggplot2 version 3.3.2 library [69]. To better understand the differences between the initial and final stages of both sexual and apomictic plants, ORA results were filtered by specificity to a particular stage.

## 3. Results

### 3.1. Gene Expression

The assembled transcriptome generated a total of 162520 trinity unigenes with a 43% GC content and a contig N50 of 2128 (Appendix A) According to BUSCO, 90% completeness was achieved in the de novo assembled transcriptome, indicating that we have generated a high-quality transcriptome assembly that could be used for further downstream analysis (Appendix A). The total number of expressed unigenes among *Limonium* samples was highest in ovules from *L. multiflorum* (apomictic) in stage S2 (115775), followed by *L. dodartii* (facultative apomictic) in stage S4 (103345), and varying from 20133 to 76395 in sexual plants (Table 2). Among these, the number of expressed unigenes in *L. auriculifolium* was higher than that in *L. ovalifolium* in stages S3/S4, but lower in the remaining stages (Table 2). In the PCA, PC1, which accounted for 71% of the variance, revealed a clear cluster of sexual plants on the right side of the graph (Appendix A). The PC2 separated sexual ovules in the S1 and S2 stages (upper-right quadrant) from sexual ovules in the S3/S4 stage (lower-right quadrant). Moreover, PC1 grouped all samples from apomictic and facultative apomictic plants with ovules in stage S1 (Appendix A, left), showing a higher dispersion in the remaining stages of these ovules.

Several HKGs were found in all comparisons of *Limonium* plants, regardless of species, reproductive strategy or stage, namely: ACT domain-containing proteins (*ACR3, ACR8,* and *ACR9*), actin (*ACT2, ACT4,* and *ACT7*), actin-depolymerizing factors (*ADF1* and *ADF10*), actin-interacting proteins (*AIP1-1, AIP1-2*), actin-related proteins (*ARP2, ARP6,* and *ARP8*), cytosolic Fe-S cluster assembly factor *NBP35*, expansins (*EXPA2* and *EXPA9*), glyceraldehyde-3-phosphate dehydrogenase *GAPA1,* heat shock proteins (*HSP90-4* and *HSP90-5*), NADH dehydrogenase [ubiquinone] flavoprotein 2 (*NDUFV2*), phosphoglycerate kinase 1 (*PGK1*), polyubiquitin 3 (*UBQ3*), TATA-box-binding protein 2 (*TBP2*), tubby-related proteins (*TULP1* and *TULP3*) and tubulins (*TUB1, TUBB1, TUBB4,* and *TUBB7*).

### 3.2. Overall Differential Expression Analysis

The highest number of annotated DEGs (3544) was found in M2o4, followed by M2a4 (1174) (Table 1). In intraspecies comparisons, *L. ovalifolium* showed the lowest number of DEGs in O2o1 (693), followed by O4o2 (1650), with the major differences being observed in O4o1 (2067; Appendix A). Conversely, *L. auriculifolium* showed the lowest differences in A4a1 (351), followed by A2A1 (796), thus presenting the major differences in A4a2 (1346; Appendix A). Notably, DEGs showed a proportion of 24% to 35% unannotated genes across samples (Table 1). On the other hand, M1o1 and M1o2 showed just slightly more DEGs than M1a2, while M2m1 presented more DEGs than M2d4. When comparing the two sexual plants, there was an increasing number of DEGs with the progression of stages, which varied from 616 to 1611 (Table 1; Appendix A).

When comparing DEGs between apomictic and sexual species, among sexual plants in S1, 47% (602) were found both in M1a1 and M1o1, while in S2, 32% (407) were found both in M1a2 and M1o2 (Figure 1A,B). The number of specific DEGs was higher in M1a1 than in M1o1 (478 vs. 207), but lower in M1a2 than in M1o2 (256 vs. 611). Moreover, the number of specific DEGs was much higher in M2o4 than in M2a4 (2562 vs. 272), with the lowest in M2d4 (71; Figure 1C). 

Some DEGs were detected in more than one developmental phase, although they presented opposite regulations (Appendix A; Appendix A). Only four DEGs were down-regulated in M1a1 and up-regulated in M1o1, including FPF1, associated with flower development. Furthermore, 124 DEGs were down-regulated in M2a4, including *ABCG28*, *ADPG2*, *AGL66*, *ATXR5*, *CALS5, CNGC18*, *LRL1*, *PRK3*, *PS1*, *SHT*, *TIP5-1*, *WRKY2* associated to anther and pollen development, but up-regulated in M2o4, while 5 DEGs presented the opposite regulation, namely *SEU*, which is involved in flower development. Additionally, 11 DEG were up-regulated in M2a4 but down-regulated in M2d4, being mainly linked to stress responses. Moreover, 44 DEGs were up-regulated in M2o4 but down-regulated in M2d4, which included QRT2, related to anther and pollen.

### 3.3. DEGs Potentially Implicated in Apomixis Regulation

Some common HKGs in *A. thaliana* were found to be differentially expressed in apomictic *Limonium* plants, namely *A1*, *ACR11,* two *ACT,* seven *EXPA,* two *GAPC, HSP90-6, RALFL19,* four *TUBB*, and eight *UBC* genes. While *EXPA* and *TUBB* DEGs were mainly down-regulated in apomictic plants, the remaining genes were mostly up-regulated (Table 3).

Oxidative stress-related DEGs were found to be mainly up-regulated in apomictic plants, namely *ACO3*, *CDSP32*, *GSH1*, *GSTU20*, *ABC1K8*, and *APX6* (Appendix A). However, *GR1*, *GASA14* (GASA—GA-stimulated transcripts), and *GSTF11*, which are also related to oxidative stress, were down-regulated. Some genes presented mixed regulation, with *GSTU19* being down-regulated in M1o2 but up-regulated in M1a2 and M2o4, and *MIOX* being up-regulated in all comparisons except M2d4, which was down-regulated (Appendix A).

Analyzing DEGs associated with sRNA biogenesis showed an up-regulation of various genes like *AGO1*, *AGO4*, *AGO5*, *AGO7*, *AGO8*, *AGO9*, *DML2*, and *DNMT2* in M2o4. Nevertheless, in the remaining comparisons between apomictic and sexual ovules, while there was a down-regulation of *AGO5* and *AGO10* in apomictic ovules, DML2 was up-regulated in the same plants (Appendix A).

### 3.4. Floral-Related DEGs

Globally, the most differentially expressed genes between apomictic and sexual ovules in the early stages were associated with floral development (Appendix A). In M1a1, top-DEGs included a down-regulation of *AP3*, *PI*, and *PEX4* (log2FC: −8.66, −8.50, and −8.44), while *AP1* and *SEP1* were among the top down-regulated DEGs in M1a2 (log2FC: −7.24 and −7.09) (Appendix A). In M1o1, top down-regulated DEGs included *SEP1* and *GASA7* (log2FC: −7.56 and −7.43), while *AP3* was down-regulated in top-DEGs in M1o2 (log2FC: −8.55) (Appendix A). Overall, top-DEGs from M2a4, M2o4, and M2d4 were implicated in general molecular functions, such as binding, a structural constituent of the ribosome, and catalytic, transporter, structural molecule, ATP-dependent, and transcription regulator activities, presenting a higher log2FC variation in M2a4 (−8.30 to 12.35) and M2o4 (−8.52 to 16.87) than in M2d4 (−5.91 to 8.57) (Appendix A). Moreover, among top-DEGs in M2m1, two genes presented opposite regulation, with *GASA1* being down-regulated and *AGL15* being up-regulated (log2FC: −4.61 to 8.81) (Appendix A). In the remaining comparisons, there was also a predominance of general molecular functions (Appendix A).

In the early stages, among all DEGs between apomictic and sexual ovules, there was a predominant down-regulation of *AP*, *PI*, and *SEP* in apomictic plants. Additionally, it was found that AGAMOUS (AG) genes (e.g., *AGL42*) were up-regulated, as were other MADS-box genes, namely *ANR1*, *FLC*, *SOC1*, and *SVP*. However, M2o4 showed both up- and down-regulation of AP genes. Other MADS-box genes were also both up- and down-regulated in M2a4 and M2o4 (Figure 2 and Table 4).

In all comparisons, TFs potentially related to male sterility were mostly associated with down-regulated DEGs (Figure 3). These TFs were classified into 10 major families, namely *AP2/ERF*, *bHLH*, *bZIP*, *C2C2*, *C2H2*, *HB*, *MADS*, *MYB*, *NAC*, and *WRKY*, from which the most representative families in all comparisons were *WRKY* and *MYB*. These TFs were particularly abundant in DEGs in M2o4 and particularly low in M2d4. Furthermore, TFs from apomictic DEGs showed an enrichment of metabolic pathways in the KEGG database. “Plant hormone signal transduction” (KEGG:04075) was up-regulated in M1o1 and M2o4, showing both up- and down-regulation in M1o2 and in M2a4, involving two *AHK*, *ARF1*, two *ARR*, *BZR2*, *DPBF3*, *EIN4*, *ERF2*, *ETR1*, *GBF4*, four *IAA*, *MYC2*, *NPR5* and three *TGA*. Up-regulation of “Lysine degradation” (KEGG:00310) and “MAPK signaling pathway—plant” (KEGG:04016) were only enriched in M2o4. The first was associated with *ASHR1*, two *ATXR*, *EZA1*, two *SUVH* and *SUVR3*, while the second involved *EIN4*, *ETR1*, *MYC2* and two *WRKY*. Moreover, WikiPathways “flower development” (WP:WP618) and “flower development (initiation)” (WP:WP2108) were enriched among up-DEGs across comparisons, being related to AG, three *AP*, *PI*, *RAP2-7*, two *SEP*, *SOC1*, and *SVP* (Appendix A).

Additionally, four DEGs were found to be associated with tryptophan metabolism, namely *TAA1*, *TSB2*, and AT3G04600, which were up-regulated in M2o4 and TAR2, which was down-regulated in most comparisons (Table 5). Moreover, 29 DEGs were found to be related to ethylene, namely *AIL5*, *ANT*, *CRF2*, *EIN2*, 14 *ERF*, *ERS1*, five *RAP2*, *RTE1*, *SHN3*, *TINY*, *WR11*, and AT4G13040, the majority of which were up-regulated in apomictic plants, especially in M2o4. Furthermore, 12 DEGs were found to be related to aminoacyl-tRNA, namely *AO*, *EDD1*, *GDH2*, two *GLDP*, *GRDP2*, *PSS1*, three *RBG*, *RZ1A*, and *UGLYAH*, which showed a similar regulation. Furthermore, 29 DEGs were found to be related to lysine, namely *AATL1*, *ASHR1*, three *ATX*, four *ATXR*, *ELF6*, *EMB3003*, *EZA1*, three *JMJ*, two *LHT*, *LTA2*, *OVA5*, two *SUVH*, three *SUVR*, AT1G25530, AT4G26910, AT5G55070, AT4G35180, and AT3G11710, presenting mostly up-regulation in apomictic plants (Table 5).

### 3.5. General GO Enrichment

DEGs between apomictic and sexual ovules in early stages were found to be enriched in seven main ancestor terms (Figure 4A). Most biological regulation terms (GO:0032501, GO:0048580, GO:2000241) were mainly enriched in up-DEGs in M1o2, although “regulation of flower development” (GO:0009909) was enriched in down-DEGs (Figure 4A). Cellular processes (GO:0007015, GO:0030029) were enriched among down-DEGs of M1o1 and M1o2. Although some developmental processes (GO:0009653, GO:0010623) were enriched among down-DEGs, “flower development” (GO:0009908) was enriched among up-DEGs. Localization terms related to biomolecules (GO:0015800, GO:0008643) were down-regulated, while transmembrane transports (GO:1903959, GO:0055085) were up-regulated in M1a1 and M1a2, but mainly down-regulated in M1o1 and M1o2. Although several metabolic processes (GO:0006631, GO:0045490, GO:0009699, GO:0000272, GO:0006468) were down-regulated, photosynthesis-related processes (GO:0005996, GO:0015979, GO:0019684) were up-regulated. Most responses to stimuli (GO:0009416, GO:0014070, GO:0006979) were enriched in up-DEGs. Finally, signaling terms (GO:0023052, GO:0007166) were mainly up-regulated (Figure 4A).

DEGs from M2a4 and M2o4 showed many enriched terms, especially in the up-DEGs of the latter. DEGs from M2d4 presented only six enriched metabolic process terms associated with down-regulated DEGs (Figure 4B). Most biological regulation terms (GO:0010629, GO:0048522, GO:0010646, GO:0009966, GO:0023051, GO:0010119) were enriched among up-regulated DEGs from M2o4, while “regulation of pollen tube growth” (GO:0080092) was down-regulated in M2a4. Although most cellular process terms were up-regulated in M2o4 (GO:0007049, GO:0051301, GO:0071840, GO:0007059, GO:0051321, GO:0090332, GO:0010118), they were down-regulated in M2a4 (GO:0071840, GO:0032501, GO:0009826). “Callose localization” (GO:0052545) was only enriched for down-DEGs in M2o4. Reproductive processes presented both types of regulation, where “pollination” (GO:0009856) and “pollen tube development” (GO:0080092) were down-regulated in M2o4 and M2a4, respectively, but “reproductive structure development” (GO:0048868) was enriched among up-DEGs in M2a4. Metabolic process and response to stimulus terms were mainly enriched in up-DEGs from apomictic ovules relative to both sexual species (Figure 4B).

### 3.6. GO Enrichment in Floral and Pollen-Related DEGs

Among all DEGs, 49 were found to be floral-related, which were mainly up-regulated for flowering and gibberellin-related terms (GO:0009908, GO:0010228, GO:0009685, GO:0010077, GO:0009739, GO:0048573, GO:0048437; Figure 5). Conversely, there was a predominance of down-regulation in DEGs related to brassinosteroid, ovule and inflorescence development, and floral organ identity (GO:0010268, GO:0009741, GO:0048481, GO:0010229, GO:0010093).

Overall, among DEGs annotated with pollen-related GO terms, the majority were downregulated in all comparisons (Figure 6; Table 6). However, “pollen development” (GO:0009555), “pollen maturation” (GO:0010152), “pollen tube” (GO:0090406), “pollen tube growth” (GO:0009860) and “pollen tube guidance” (GO:0010183) were up-regulated in M2o4. Globally, both up- and down-regulated DEGs were especially related to pollen development. Noticeably, down-regulated DEGs were also related to pollen wall assembly and pollen tube development and growth. Specifically, the *EX5* gene was found to be down-regulated in M1a1 and M1o2.

## 4. Discussion

An increasing number of molecular studies have identified several candidate genes implicated in the shift from sexual to apomixis reproduction [18,21,22,70]. In different species, apomixis has been found to arise due to the action or deregulation of different genes associated with the normal sexual pathway [23,71,72,73]. Nevertheless, it is still not fully understood how these genes alter reproductive pathways to establish apomixis. 

In most apomictic wild species. Implementing omics approaches can be particularly challenging as complete genomic sequences are not available and, therefore, genome annotation information is not available. Additionally, obtaining plant material for transcriptomic studies can be an experimentally challenging task in *Limonium* since each plant presents a single ovary, enclosed in a calyx and inner, medium, and outer bracts that yield just a single basal ovule [13]. In the current study, we performed a comparative transcriptome analysis between sexual and asexual plants and identified candidate genes that are specifically or differentially expressed between reproductive modes and among stages of ovule development. This approach allowed us to disclose differential regulation of both HKGs as well as genes specifically involved in flower development, male sterility, and pollen recognition, besides major pathways potentially central to apomixis, including protein degradation, transcription, stress response, hormonal signaling, signal transduction, and epigenetic regulation.

### 4.1. Differential Regulation of HKG and Metabolic Pathways in Sexual and Apomictic Plants

In this study, the total number of expressed unigenes among samples was higher in ovules from apomictic plants than in those from sexual plants, together with a differential regulation of genes, particularly in the later stages of ovule development (Table 1). Previous studies between sexual and asexual plants provided support for the deregulation of reproductive pathways, including HKGs in, e.g., *Boechera holboellii* complex [72], *Brachiaria* [74], *Cenchrus ciliaris* [75], and *Ranunculus* [73], among others. In this study, although many homolog genes in *Arabidopsis* were stably expressed in *Limonium*, such as *ACT* domain-containing proteins. Cytosolic Fe-S cluster assembly factors *NBP35*, NADH dehydrogenase [ubiquinone] flavoprotein 2, phosphoglycerate kinase 1, polyubiquitin 3, TATA-box binding protein 2, and other HKGs were found to be differentially expressed. These include genes related to the ubiquitin degradation process, such as ubiquitin-conjugating enzymes, tubulin, actin, and elongation factor-1 α as found in other sexual and apomictic plants’ complexes above referred. Therefore, some of the HKGs identified (Table 3) in our study can be potentially used as reference genes to be validated in future quantitative gene expression studies using different developmental stages of specific tissue types or different reproductive modes.

A differential representation of DEGs in *Limonium* sexual and apomictic plants associated with the oxidative stress response was found. In apomictic plants, some of these DEGs (Appendix A), e.g., *ACO3*, *CDSP32*, *GSH1*, etc., were up-regulated while others were down-regulated (*GR1*, *GASA14*, and *GSTF11*) in both the initial (apomeiosis) and later (parthenogenesis) stages of ovule development. Nonetheless, apomicts present more up-regulated DEGs regarding oxidative stress than sexual plants, supporting the involvement of redox reactions in this reproductive mode. Alteration of homeostasis-based processes of stress perception and attenuation in sexual species of several genera would induce apomeiotic spores and gametophyte formation [23]. Apomeiosis occurs when the redox balance is more toward H_2_O_2_ catabolism, and the transition from meiosis to apomeiosis can be changed by a disturbance in this homeostasis [23]. 

Among DEGs between sexual and *Limonium* apomicts, most were up-regulated in the latter stages of ovule development (parthenogenesis), such as *ATRX* genes coding for chromatin remodelling proteins as well as multiple histone methylation genes concerning epigenetic developmental mechanisms in plants [76] (Table 5 and Table 6). These DEGs were also previously found to be upregulated, for instance, in parthenogenetic eggs of *Cenchrus ciliaris* [77]. Moreover, other DEGs implicated in small RNA biogenesis and DNA-methylation pathways, such as the *AGO9* and *AGO4* homolog genes in *Arabidopsis* mutants found to be associated with phenotypes reminiscent of apospory or diplospory [78], were also detected in our study. In *Boechera* apomicts, *AGO9* was found at low levels in the megaspore mother cell itself, becoming an apomictic initial cell [79]. However, in our study, *AGO4* and *AGO9* seem to have more specific roles in ovules at later stages of development (parthenogenesis), likely being involved in eggs assuming a parthenogenesis fate. MAPK signaling and aminoacyl-tRNA biosynthesis pathways that perform roles in translational regulation, RNA splicing, and tRNA proofreading [80] also showed transcriptional changes at this stage in apomictic *Limonium* plants (Table 5 and Table 6).

DEGs were also remarkably enriched in genes implicated in hormonal signaling, such as the ethylene signaling pathway, in which the apomictic gametophytes overexpressed 26 ethylene-responsive transcription factors (Table 5 and Table 6). For example, in *Cenchrus ciliaris*, EIN2 (ethylene insensitive 2) together with 14 ethylene responsive transcription factors were up-regulated in parthenogenetic eggs [77], although in our study both genes showed contrasting expression patterns in the same developmental stage (parthenogenesis). Moreover, in the parthenogenetic ovules, overexpressed genes were related to tryptophan metabolism, such as *TAA1* (l-tryptophan pyruvate aminotransferase), which converts tryptophan to indole-pyruvic acid, a direct biosynthetic precursor of the auxin in *Arabidopsis* (IAA [81]). Crosstalk between ethylene signaling and auxin pathways is involved in the regulation of developmental processes [82]. 

### 4.2. Feminization of Apomicts Is Related to Down-Regulation of Floral Genes Specifying Stamens 

Besides auxin, other hormones like gibberellic acid (GA) contribute to flower development, the development of male and female gametophytes, and seed germination [83.84]. The GASA genes as well as the GA biosynthesis genes in *Arabidopsis*, implicated in controlling floral induction, seed maturation, and germination [83,84], were differentially expressed between apomictic and sexual plants in our study (Table 4). One of the targets of GA signaling are the floral homeotic genes encoding MADS-box transcription factors involved in floral development in accordance with the ABCDE model [85]. In our study, among the top DEGs between sexual and apomictic plants and between the different ovule stages, MADS-box transcription factors were identified, including floral homeotic genes with a MADS-box domain. In *A. thaliana*, the MADS-box from A-class genes (*AP1*; *AP2*) specifies the formation of sepals; the combination of A- and B-class genes (*AP3*; *PI*) determines petal’ development; the B-, C- (*AGL*), and E-class (*SEP*) genes specify stamens; and the C- and E-class genes specify carpels. Only the expression of genes from class C specifies carpel formation. Class E genes (*SEP3*) are associated with the formation of all flower whorls. The gene classes A and C are expressed antagonistically; the A gene class is expressed in sepals and petals, and the C gene class is expressed in stamens and carpels [86,87]. 

GA promotes reproductive development by upregulating expression of the floral meristem identity gene LEAFY (*LFY*), which in turn upregulates expression of *AP3* and *AGL* that, in conjunction with *PI* and *SEP3*, regulate floral organ identity [87]. In our study, we found changes in the expression of MADS-box genes in the different stages of ovule development between sexual and apomictic plants, particularly *PI*, *SEP1*, and *AP3* genes, which were downregulated in apomicts. The *PI/AP3* genes have a role in sexual dimorphism and have been identified as masculinizing factors in spinach [88]. In dioecious plants such as *Populus*, constitutive overexpression of *PI/AP3* produces male flowers, but in female flowers the presence of a feminizing factor F downregulates *PI/AP3*, diverting development to a female developmental pathway, inhibiting stamens, and allowing carpels to form [89]. Therefore, it could be hypothesized that male sterile *Limonium* plants with homeotic changes in floral organs lack the gene function of the corresponding class B genes.

### 4.3. Male Sterility Appears to Be Linked with Downregulation of Genes Connected to Pollen Wall Formation and Assembly and Pollen Tube Growth 

Various genes are involved in pollen wall development and assembly, which is a specialized extracellular cell wall matrix that encases the male gametophytes [90]. A specific cell wall polymer also known as β-glucan is synthesized by callose synthases in pollen mother cells and microspore tetrads that acts as a template for primexine, thus providing a structural basis for exine formation [90]. Callose synthase 5 (*CSL5*) is a key isoform of callose synthases responsible for the formation of the callose wall, which is essential for the accumulation of callose in the tube wall and the callose plug in growing pollen tubes [91]. In *CSL5* mutants, the viability of pollen grains is greatly reduced in *Arabidopsis* [91,92] and rice [93]. In our study in *Limonium, CSL5* is downregulated in the initial (apomeiosis) and later phases of development (parthenogenesis) in the apomicts. One of the characteristic features of plants in the *L. binervosum* group is the widespread existence of male sterility, i.e., a lack of pollen [12,13,26,36]. Electron microscopy studies showed that *L. multiflorum* plants had many flowers with empty anthers and sometimes flowers with no pollen at all; the few microspores that formed showed collapsed morphology and lacked the typical exine patterns [12]. In *L. multiflorum* apomicts, after anther dehiscence releases pollen, the plants never undergo their first mitosis, only attaining the ‘‘ring vacuolate’’ stage, and the male germ unit is not produced [12]. Interestingly, in this study, callose synthase isoforms such as *CSL9* or *CSL11* that were upregulated in the apomicts have a role in *Arabidopsis* pollen mitosis by disrupting pollen mitosis and producing pollen with only one or two nuclei, the generative cell being degenerated, undifferentiated, or mislocalized [94,95], as found here in *Limonium*.

Moreover, in our study, the gene *PEX4* that codes for extracellular glycoproteins that belong to the hydroxyproline-rich glycoprotein family and have a role in pollen germination and pollen tube growth in *A. thaliana* [96,97]. was downregulated in *Limonium* apomicts (apomeiosis, M1a1). The *PEX4* mutants have an excessive deposition of callose [96,97], leading to abnormal pollen tubes that develop bulges and burst [97]. While in *L. multiflorum* apomicts, pollen tubes are never observed since unicellular pollen never undergoes its first mitosis. In *L. ovalifolium* sexual plants, pollen grains follow a first asymmetric mitotic division, producing a generative cell within the vegetative pollen grain cell in the binucleate pollen stage [12]. Therefore, our results support a role for *PEX4* in pollen tube growth. Perhaps this gene, along with other unknown genes, might create a terminal combination that does not allow the development of pollen tubes.

### 4.4. Pollen-Stigma Interactions

Pollen-pistil interactions can be viewed as a major prezygotic pollination reproductive barrier and are active systems of pollen rejection [98]. Some of these systems act at the level of the stigma, with a genetic control independent from embryo sac development involving S-alleles [99]. In the Brassicaceae family, the SI is controlled sporophytically by a single S locus that incorporates stigma-expressed and anther-expressed genes composed of multiple alleles or variants [100]. In various gametophytic apomicts, non-functional pollen can cause a weakening of SI and a breakdown of the sporophytic SI system (mentor effects [9]). 

*Limonium* species show a polymorphic sexual system associated with flower polymorphisms and a sporophytic self-incompatibility that prevents self- and intramorph mating [29.30]. *Limonium* gametophytic apomicts that form diplosporic (apomictic) embryo sacs of *Rudbeckia* type like in *L. multiflorum* [13] show abnormal and non-functional pollen due to a sporophytic defect [12]. In our study, in the GO term “recognition of pollen” (GO:0048544; Table 6). All DEGs were lectin receptor kinases (LECRKs) [101], except for AT3G49500. These kinases belong to the class of G-LECRKs [101], particularly the S-locus Receptor Kinase (SRK), known for its role in self-incompatibility [101] and potentially of high interest in our studied species. In our study, DEGs from this LECRKs complex were detected under the GO term “recognition of pollen”, namely AT1G61380, AT1G65800, AT5G24080, AT4G27300, AT4G21380, and AT4G27290. Four DEGs of the SRK were overexpressed, namely in the initial stages of ovule development (apomeiosis), such as AT1G61380 (M1a1 and M1a2) and AT1G65800 (M1a1), as well as in later ovule stages (M2a4 and M2o4), AT4G27300 (M2o4), and AT4G27290 (M2o4). These findings indicate that the genes implicated in pollen recognition in *Limonium* were already expressed at earlier ovule stages. This implies that the fate of the gametophytic apomict male spores is decided by the maternal genes in the initial ovule stages. Nonetheless, a genetic linkage between SI and apomixis cannot be easily assumed, since the breakdown of SI mostly affects the pollen and the stigma, while apomixis affects the development of the embryo sac. 

### 4.5. Up-Regulation of Specific Genes Related with Embryo Formation in the Apomicts

In *Arabidopsis,* flowering can be promoted by repressing the transcription of the central flowering repressor and vernalization regulatory gene *FLC* (FLOWERING LOCUS C), which belongs to the MADS-box class of transcription factors [102]. The *FLC* seems to regulate several transcription factors involved in important biological processes such as reproductive and embryonic development [103]. *FLC* impedes the floral transition by inhibiting the expression of the floral primordium identity genes such as *FT* (FLOWERING LOCUS T), *SOC1/AGL20* (SUPPRESSOR OF CONSTANS OVEREXPRESSION 1/AG20), *LFY*, *AP1*, and floral organ identity *AG* and *AP3* genes [104]. Another gene involved in the genetic control of flowering time in *Arabidopsis* is *FPF1* (FLOWERING PROMOTING FACTOR), which modulates the acquisition of competence to flower in the apical meristem and is expressed earlier than *AP1* [105]. In our study, the *FPF1* and *SOC1* showed higher levels of regulation in apomicts than in sexual plants, whereas *AP1* showed reduced levels in the first ones. These results indicate differences in the regulation of major genes controlling the transition from a vegetative to a reproductive mode in the apomict’s apical meristem. Interestingly, in our study, both *FLC*, *AGL6*, and *AGL15*, as well as other MADS-box transcription factors, were specifically upregulated in the later stages of development. In *A*. *thaliana AGL6* functions in the early stages of the flowering signal transduction pathway by inhibiting the transcription of *FLC* genes [106]. In *Brachiaria brizantha*, *AGL6* is differentially expressed during embryo sac formation in apomictic and sexual plants [107]. In our study, *AGL15*, which plays an essential role during early zygotic embryogenesis in *Arabidopsis* [108], in *Brassica napus,* and in soyabean somatic embryos [109], was specifically upregulated in the later phase of ovule development in *Limonium* apomicts. The *AGL15* is a component of the SERK protein complex [110] that is part of a molecular network linked to zygotic and somatic embryogenesis. However, in this study, we were unable to find any differential expression of SERK-like genes. 

Some genes related to the bioactive gibberellins’ deactivation reaction from the GA2OX family were differentially expressed. For example, *GA2OX6* was up-regulated in both the initial and later stages of ovule development. While in *A. thaliana, GA2OX6* expression is activated by *AGL15* during embryogenesis [111], in our study, *AGL15* was down-regulated in M2a4 and up-regulated in M2o4, suggesting differences among species. Moreover, *GA2OX6* is found to be expressed in sepals, stigmas, and immature anthers. Regarding seed development, it was only expressed in the antipodal cells before the 8-cell stage, suggesting that this gene is a negative regulator of seed germination [111]. Remarkably, from the same family, *GA2OX8*, which is exclusively expressed in stomatal cells in *A. thaliana* but is not expressed or has distinct expression patterns in flower tissues or seed development [111], was down-regulated in *Limonium* apomicts in later stages of ovule development (parthenogenesis). This finding supports the idea that this gene can have other or different roles in *Limonium*.

## 5. Conclusions

This study sheds light on genes involved in *Limonium* sexual and apomictic reproduction. The findings substantiate the deregulation of gene expression in the regular sexual pathway. While several HKGs are found to be differentially expressed between sexual and asexual plants, other genes are found to be stably expressed. These can be potentially used as reference genes to be validated for specific tissue types (vegetative and reproductive) or reproductive modes (sexual and apomictic) in future quantitative gene expression studies. Our findings reveal that the latter stage of ovule development (parthenogenesis) was the most contrasting phase in terms of differential gene expression between asexual and sexual plants. Among them, the MADS-box domain TFs are central players in many developmental processes, including control of flowering time, homeotic regulation of floral organogenesis, fruit development, and seed pigmentation.

Since *L. multiflorum* male sterile plants form parthenogenetic ovule sacs, it could be interesting to analyze candidate genes such as *PEX4* in pollen tube development and the function of AGL genes (e.g., *AGL6*) specifically modulated in the latter stages of development (parthenogenesis). Nonetheless, given the high number of 71% unannotated genes in *Limonium*, other studies are required to clarify the regulatory roles of these genes.

## Figures and Tables

**Figure 1 genes-14-00901-f001:**
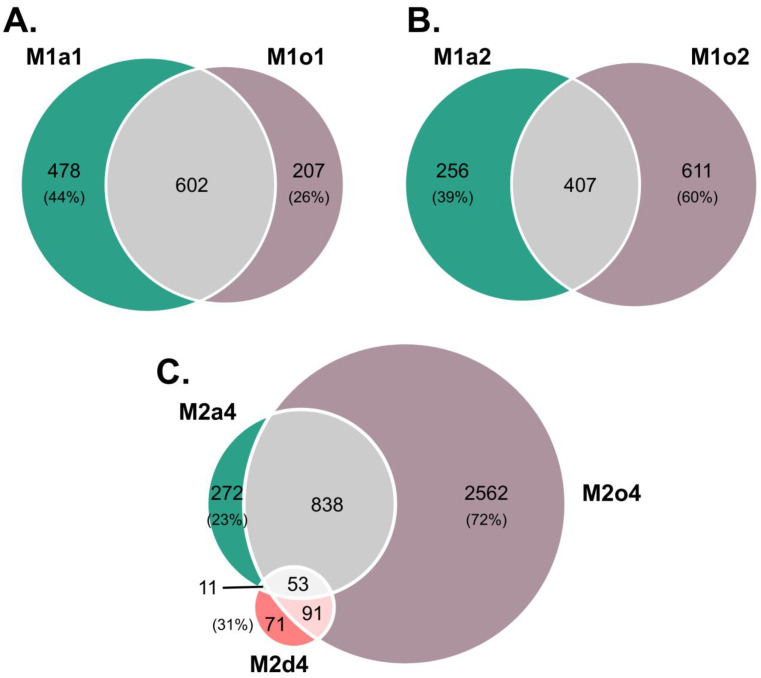
Weighted Venn diagrams of specific and overlapping differentially expressed genes (DEGs) found in the ovules of apomictic *L. multiflorum* (M), facultative apomictic *L. dodartii* (d), and sexual *L. auriculifolium* (a) and *L. ovalifolium* (o). DEGs were filtered by |log2 fold-change (log2FC)| > 2. Number of overlapping and specific DEGs in: A. *L. multiflorum* in S1 relative to *L. auriculifolium* in S1 (M1a1; green) and to *L. ovalifolium* in stage S1 (M1o1; purple); B. *L. multiflorum* in S1 relative to *L. auriculifolium* in S2 (M2a2; green) and to *L. ovalifolium* in S2 (M1o2; purple); C. *L. multiflorum* in S2 relative to *L. auriculifolium* in S3/S4 (M2a4; green), to *L. ovalifolium* (M2o4; purple) and to *L. dodartii* in S4 (M2d4; red).

**Figure 2 genes-14-00901-f002:**
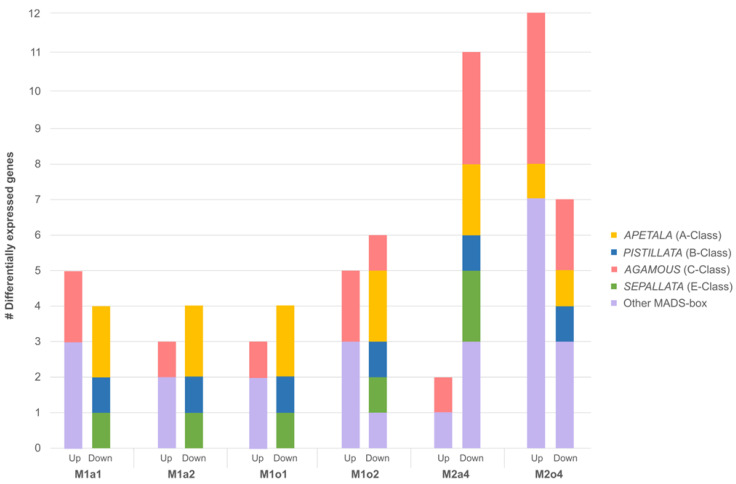
Distribution of differentially expressed genes related to floral development in ovules from apomictic *L. multiflorum* (M), sexual *L. auriculifolium* (a) and *L. ovalifolium* (o), and facultative apomictic *L. dodartii* (d). Differentially expressed MADS-box genes *APETALA* (A-class), *PISTILLATA* (B-class), *AGAMOUS* (C-class), *SEPALLATA* (E-class), and other MADS-box genes are represented. DEGs were found in apomictic ovules in S1 relative to sexual ovules in S1 (M1a1 and M1o1) and S2 (M1a2 and M1o2), and relative to sexual ovules in S3/S4 (M2a4 and M2o4), and facultative apomictic in S4 (M2d4).

**Figure 3 genes-14-00901-f003:**
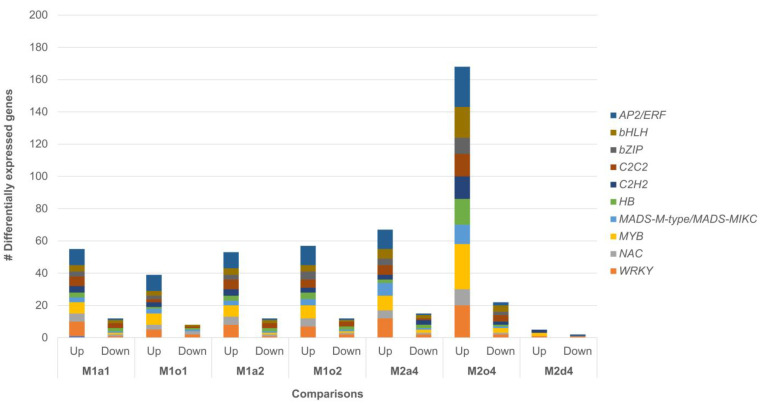
Distribution of differentially expressed transcription factors potentially related to male sterility is classified into the 10 families with the highest number of differentially expressed genes (DEGs) in ovules: Apomictic *L. multiflorum* (M), sexual *L. auriculifolium* (a) and *L. ovalifolium* (o), and facultative apomictic *L. dodartii*: *AP2/ERF*, *bHLH*, *bZIP*, *C2C2*, *C2H2*, *HB*, *MADS*, *MYB*, *NAC*, and *WRKY*.

**Figure 4 genes-14-00901-f004:**
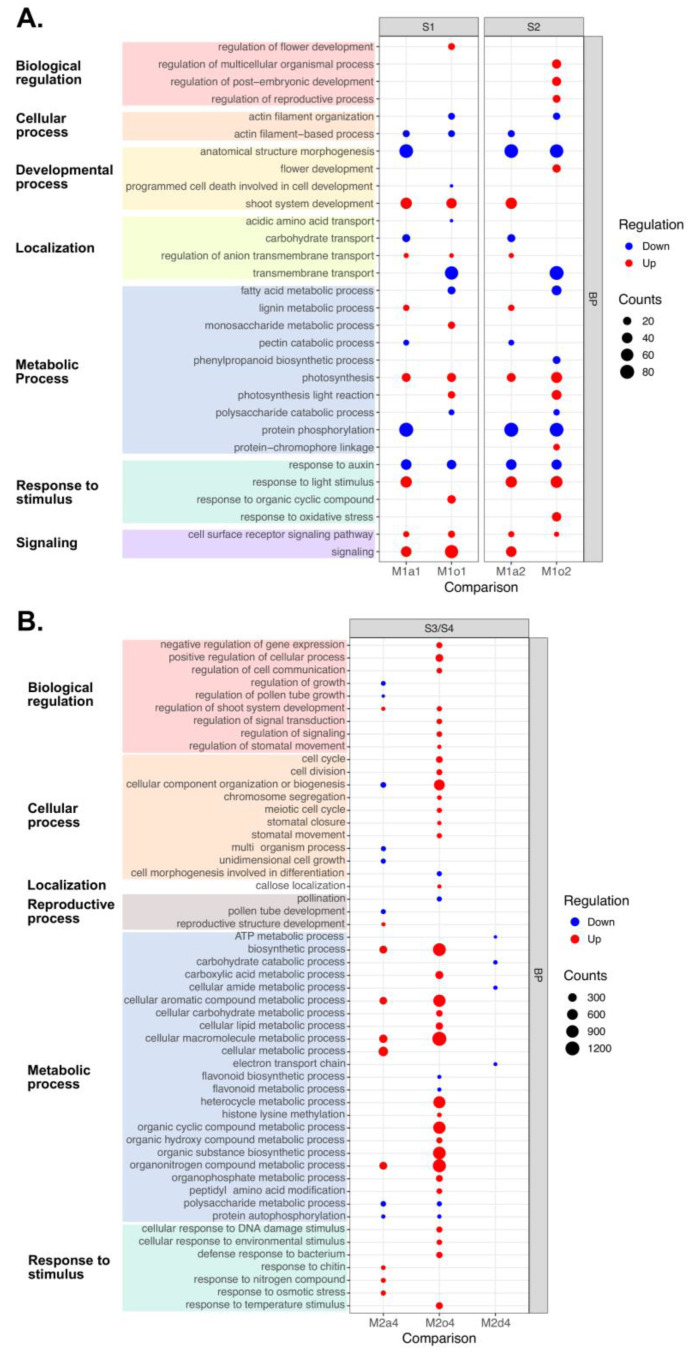
Over-representation analysis (ORA) performed by gProfiler of differentially expressed genes (DEGs) in ovules from apomictic *L. multiflorum* (M), sexual *L. auriculifolium* (a) and *L. ovalifolium* (o), and facultative apomictic *L. dodartii*. DEGs were filtered by |log2 fold-change (log2FC)| >2. *A.s thaliana,* the most similar homolog of each differentially expressed gene (DEG) was mapped to the respective functional annotation, and enriched terms were summarized using REVIGO. Significantly (FDR < 0.01), gene ontology (GO) and biological processes (BP) terms are among DEGs from (**A**) apomictic in S1 relative to sexual ovules in S1 (M1a1 and M1o1) or S2 (M1a2 and M1o2), and from (**B**) apomictic in S2 relative to sexual ovules in S3/S4 (M2a4 and M2o4), and facultative apomictic in S4 (M2d4). The dot’s size indicates the number of DEGs annotated with each term (counts), and the color shows the differential expression (red: up-regulated; blue: down-regulated). Enriched terms are grouped by their respective ancestors (ontology level 2).

**Figure 5 genes-14-00901-f005:**
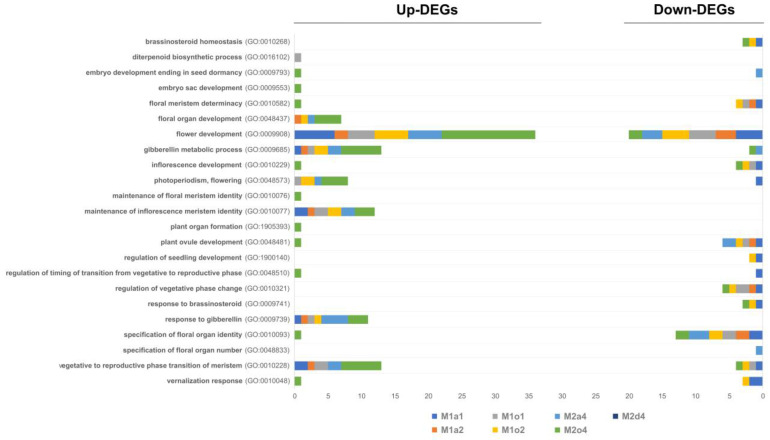
Regulation of up- and down-regulated differentially expressed genes (DEGs) in ovules in stages S1 (1), S2 (2), and S3/S4 (4) from apomictic *L. multiflorum* (M), sexual *L auriculifolium* (a) and *L. ovalifolium* (o), and facultative apomictic *L. dodartii* (d), annotated with female-related Gene Ontology (GO) terms. DEGs represent the number of significant genes found to be differently expressed in each comparison.

**Figure 6 genes-14-00901-f006:**
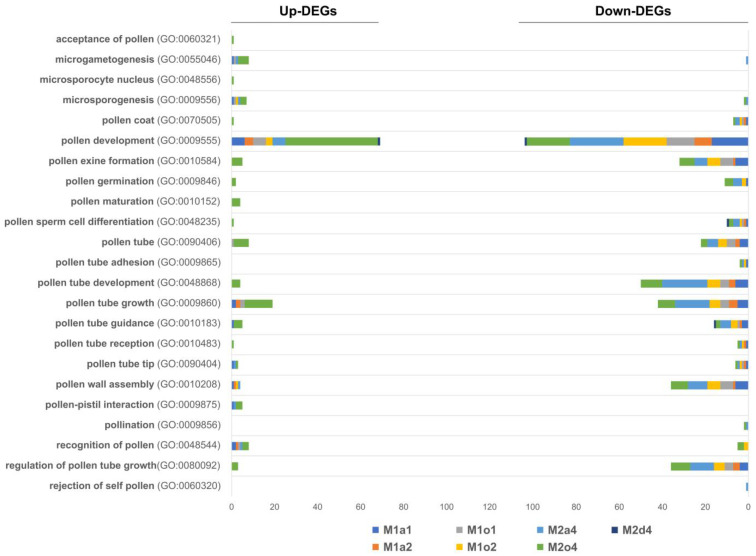
Regulation of up- and down-regulated differentially expressed genes (DEGs) in ovules in stages S1 (1), S2 (2), and S3/S4 (4) from apomictic *L. multiflorum* (M), sexual *L auriculifolium* (a) and *L. ovalifolium* (o), and facultative apomictic *L. dodartii* (d), annotated with Gene Ontology (GO) terms related to pollen. DEGs represent the number of significant genes found to be differently expressed in each comparison.

**Table 1 genes-14-00901-t001:** List of all tested and control samples of *Limonium* compared in the differential expression analysis, according to species, reproductive strategy, stage (S), and number of replicates (Rep). Number of all and annotated significantly differentially expressed genes (DEGs) detected by edgeR in *Limonium* plants, namely apomictic *L. multiflorum* (M), sexual *L. auriculifolium* (A/a) and *L. ovalifolium* (O/o), and facultative apomictic *L. dodartii* (d), in either stage S1 (1), S2 (2), or S3/S4 (4). All DEGs represent the number of significant genes found to be differently expressed between each test and control sample. DEG annotation was performed according to *A. thaliana* reference genome. [Comparisons: uppercase refers to test samples, lowercase refers to control samples, and numbers refer to respective stages].

Test Samples (T)		Control Samples (c)		Comparison	All DEGs	Annotated DEGs		
Species	Stage (Rep)	Species	Stage (Rep)			Total	Up (%)	Down (%)
*L. multiflorum* (M) [Apomictic]	**Apomictic vs. sexual**							
	S1 (R1)	*L. auriculifolium* (a) [Sexual]	S1 (R1)	**M1a1**	3517	1080	453 (42%)	627 (58%)
			S2 (R1)	**M1a2**	2785	663	295 (44%)	368 (56%)
	S2 (R1-5)		S3/S4 (R1)	**M2a4**	4400	1174	489 (42%)	685 (58%)
	S1 (R1-4)	*L. ovalifolium* (o) [Sexual]	S1 (R1)	**M1o1**	3068	809	371 (46%)	438 (54%)
			S2 (R1)	**M1o2**	3054	1018	419 (41%)	599 (59%)
	S2 (R1-5)		S3/S4 (R1)	**M2o4**	12,839	3544	2827 (80%)	717 (20%)
		**Apomictic vs. facultative apomictic**						
	S2 (R1-5)	*L. dodartii* (d) [Facultative apomictic]	S4 (R1-3)	**M2d4**	806	226	41 (18%)	185 (82%)
		**Between stages comparisons (same species)**						
	S2 (R1-5)	*L. multiflorum* (M) [Apomictic]	S1 (R1-4)	**M2m1**	1096	387	379 (98%)	8 (2%)
*L. auriculifolium* (A) [Sexual]	S2 (R1)	*L. auriculifolium* (a) [Sexual]	S1 (R1)	**A2a1**	796	276	40 (14%)	236 (86%)
	S3/S4 (R1)			**A4a1**	351	111	93 (84%)	18 (16%)
			S2 (R1)	**A4a2**	1346	471	356 (76%)	115 (24%)
*L. ovalifolium* (O) [Sexual]	S2 (R1)	*L. ovalifolium* (o) [Sexual]	S1 (R1)	**O2o1**	693	208	189 (91%)	19 (9%)
	S3/S4 (R1)			**O4o1**	2067	711	395 (56%)	316 (44%)
			S2 (R1)	**O4o2**	1650	526	228 (43%)	298 (57%)
		**Between species comparisons (sexual)**						
	S1 (R1)	*L. auriculifolium* (a) [Sexual]	S1 (R1)	**O1a1**	616	200	34 (17%)	166 (83%)
	S2 (R1)		S2 (R1)	**O2a2**	1242	367	273 (74%)	94 (26%)
	S3/S4 (R1)		S3/S4 (R1)	**O4a4**	1611	550	257 (47%)	293 (53%)

**Table 2 genes-14-00901-t002:** Total number of genes expressed by *Limonium* samples from apomictic *L. multiflorum* (M), facultative apomictic *Limonium dodartii* (d), and sexual *L. auriculifolium* (A/a) and *L. ovalifolium* (O/o) ovules in stages S1, S2, and S3/S4.

Reproduction	Species	Stages		
		S1	S2	S3/S4
Apomictic	*L. multiflorum* (M)	933936	115775	-
Facultative apomictic	*L. dodartii* (d)	-	-	103,345
Sexual	*L. auriculifolium* (A/a)	60,143	48,851	61,550
	*L. ovalifolium* (O/o)	67,207	76,395	20,133

**Table 3 genes-14-00901-t003:** List of common housekeeping genes in *A. thaliana* (gene name) that were differentially expressed (DEGs) in the ovules of apomictic *L. multiflorum* (M) in S1 and S2, and sexual *L. auriculifolium* (a) and *L. ovalifolium* (o) in S1, S2, and S3/S4. DEGs were filtered by |log2 fold-change (log2FC)| >2 (red: up-regulated DEGs; blue: down-regulated DEGs).

Gene ID	Gene Name	Protein Name	Log2FC						
			M1a1	M1a2	M1o1	M1o2	M2a4	M2o4	M2d4
TRINITY_DN12562_c0_g1	*A1*	Elongation factor 1-α						10.09	
TRINITY_DN9323_c2_g1	*ACR11*	ACT domain-containing protein ACR11						7.81	
TRINITY_DN594_c3_g1	*ACT1*	Actin-1						7.09	
TRINITY_DN3478_c1_g1	*ACT11*	Actin-11					−4.39	−2.32	
TRINITY_DN13623_c0_g1	*EXPA11*	Expansin-A11					−5.35	−5.85	
TRINITY_DN2256_c1_g1	*EXPA13*	Expansin-A13	−2.28			−2.07	−2.17	−3.78	
TRINITY_DN27728_c0_g1	*EXPA15*	Expansin-A15						6,5	
TRINITY_DN151736_c0_g1	*EXPA16*	Expansin-A16					−4.77	−2.4	−2.06
TRINITY_DN12567_c0_g1	*EXPA20*	Expansin-A20					−2.08	7.23	
TRINITY_DN21891_c0_g1	*EXPA6*	Expansin-A6	−2.41			−2.42			
TRINITY_DN5240_c0_g1	*EXPA8*	Expansin-A8	−5.09	−3.65	−3.79	−4.76	−2.76	−3.16	
TRINITY_DN10654_c0_g1	*GAPC1*	Glyceraldehyde-3-phosphate dehydrogenase GAPC1, cytosolic		8.03	2.95				
TRINITY_DN10775_c0_g1	*GAPC2*	Glyceraldehyde-3-phosphate dehydrogenase GAPC2, cytosolic					−2.67	−2.02	
TRINITY_DN638_c0_g1	*HSP90-6*	Heat shock protein 90-6, mitochondrial						9.87	
TRINITY_DN28088_c0_g2	*RALFL19*	Probable ubiquitin-conjugating enzyme E2 24	−7.67	−4.99	−4.97	−7.43	−4.92	−4.37	
TRINITY_DN521_c0_g6	*TUBB5*	Tubulin β-5 chain					−2.46	−3.5	
TRINITY_DN159888_c0_g1	*TUBB6*	Tubulin β-6 chain	−2.58					−2.21	
TRINITY_DN2826_c0_g1	*TUBB8*	Tubulin β-8 chain						11.41	
TRINITY_DN2332_c0_g1	*TUBB9*	Tubulin β-9 chain						8.91	
TRINITY_DN6957_c0_g1	*UBC10*	Ubiquitin-conjugating enzyme E2 10						8.36	
TRINITY_DN193932_c0_g1	*UBC11*	Ubiquitin-conjugating enzyme E2 11						6.39	2.72
TRINITY_DN184540_c0_g1	*UBC19*	Ubiquitin-conjugating enzyme E2 19	−3.31		−3.22	−3.64			
TRINITY_DN6909_c0_g1	*UBC29*	Ubiquitin-conjugating enzyme E2 29	3.03		2.14	3.94	4.14	14.61	
TRINITY_DN2966_c0_g1	*UBC33*	Probable ubiquitin-conjugating enzyme E2 33	2.1				5.58	9.74	
TRINITY_DN43968_c0_g1	*UBC35*	Ubiquitin-conjugating enzyme E2 35					3.13	6.36	
TRINITY_DN10935_c1_g1	*UBC4*	Ubiquitin-conjugating enzyme E2 4						7.32	
TRINITY_DN10551_c0_g1	*UBC8*	Ubiquitin-conjugating enzyme E2 8	2.06	2.34				6.99	

**Table 4 genes-14-00901-t004:** List of florally differentially expressed genes (DEGs) in ovules of *Limonium* plants, namely apomictic *L. multiflorum* (M), sexual *L. auriculifolium* (a), and *L. ovalifolium* (o). DEGs were found in apomictic ovules in S1 relative to sexual ovules in S1 (M1a1 and M1o1) and S2 (M1a2 and M1o2), and in apomictic ovules in S2 relative to sexual ovules in S3/S4 (M2a4 and M2o4); (red: up-regulated DEGs; blue: down-regulated DEGs).

Gene ID	Gene Name	Protein Name	Log2FC					
			M1a1	M1a2	M1o1	M1o2	M2a4	M2o4
TRINITY_DN28291_c0_g2	*AGL15*	Agamous-like MADS-box protein AGL15					−4.08	7.12
TRINITY_DN13295_c0_g1	*AGL16*	Agamous-like MADS-box protein AGL16	2.14	2.21				9.13
TRINITY_DN610_c1_g1	*AGL42*	MADS-box protein AGL42				2.15		
TRINITY_DN2793_c0_g2	*AGL6*	Agamous-like MADS-box protein AGL6						7.9
TRINITY_DN3873_c0_g1	*AGL65*	Agamous-like MADS-box protein AGL65				−2.29	−2.62	−3.28
TRINITY_DN12180_c0_g1	*AGL66*	Agamous-like MADS-box protein AGL66					−4.72	−3.81
TRINITY_DN342_c0_g2	*AGL8*	Agamous-like MADS-box protein AGL8	4.02		2.86	4.34	2.59	6.91
TRINITY_DN72066_c0_g2	*ANR1*	MADS-box transcription factor ANR1						6.03
TRINITY_DN9027_c0_g3	*AP1*	Floral homeotic protein APETALA 1	−7.1	−7.24	−7.17	−6.32		
TRINITY_DN2754_c0_g1	*AP2*	Floral homeotic protein APETALA 2						2.14
TRINITY_DN7779_c0_g1	*AP3*	Floral homeotic protein APETALA 3	−8.66	−6.05	−6.98	−8.55	−4.17	−3.22
TRINITY_DN46144_c0_g1	*ATH1*	Homeobox protein ATH1		3				7.07
TRINITY_DN27333_c1_g1	*BLH8*	BEL1-like homeodomain protein 8						7.15
TRINITY_DN1609_c0_g1	*BRI1*	Protein BRASSINOSTEROID INSENSITIVE 1	−2.45			−2.2		
TRINITY_DN17115_c0_g1	*CCA1*	Protein CCA1						7.18
TRINITY_DN11983_c0_g1	*CDF2*	Cyclic dof factor 2						2.81
TRINITY_DN2531_c1_g2	*COL5*	Zinc finger protein CONSTANS-LIKE 5	2.15			2.71	2.27	
TRINITY_DN26110_c0_g1	*CSTF77*	Cleavage stimulation factor subunit 77						2.13
TRINITY_DN22923_c0_g1	*ELF6*	Probable lysine-specific demethylase ELF6						5.1
TRINITY_DN3347_c0_g1	*EMF2*	Polycomb group protein EMBRYONIC FLOWER 2						2.89
TRINITY_DN7896_c0_g2	*FLC*	MADS-box protein FLOWERING LOCUS C						8.63
TRINITY_DN5378_c0_g1	*FPA*	Flowering time control protein FPA						3.27
TRINITY_DN184585_c0_g1	*FPF1*	Flowering-promoting factor 1	2.72		2.94	3.9	2.94	3.58
TRINITY_DN7899_c0_g1	*FT*	Protein FLOWERING LOCUS T				2.06		2.37
TRINITY_DN344_c1_g2	*GA2*	Ent-kaur-16-ene synthase, chloroplastic			3.21			7.82
TRINITY_DN2317_c0_g2	*GA2OX6*	Gibberellin 2-β-dioxygenase 6	3.87			3.85	3.72	11.51
TRINITY_DN38605_c0_g1	*GA2OX8*	Gibberellin 2-β-dioxygenase 8						−4,06
TRINITY_DN13033_c0_g1	*GA3OX1*	Gibberellin 3-β-dioxygenase 1					2.87	8
TRINITY_DN8333_c0_g1	*GA3OX2*	Gibberellin 3-β-dioxygenase 2						11.4
TRINITY_DN17736_c0_g1	*GASA1*	Gibberellin-regulated protein 1	4.81	2.37	2.98	5.17		
TRINITY_DN10114_c0_g1	*GASA11*	Gibberellin-regulated protein 11	2.56	2.1	2.79	2.59		9.79
TRINITY_DN31284_c0_g1	*GASA14*	Gibberellin-regulated protein 14					−2.5	−4.11
TRINITY_DN185389_c0_g1	*GASA3*	Gibberellin-regulated protein 3					−5.19	−4.01
TRINITY_DN5658_c0_g1	*GASA6*	Gibberellin-regulated protein 6	−4.47	−2.13	−2.06	−4.55	−4.83	−5.73
TRINITY_DN1467_c0_g1	*GASA7*	Gibberellin-regulated protein 7	−7.76	−6.05	−7.43	−7.99	−4.99	−4.24
TRINITY_DN8938_c0_g1	*GASA9*	Gibberellin-regulated protein 9		−2.47	−2.33			−2.99
TRINITY_DN21137_c0_g1	*GID1C*	Gibberellin receptor GID1C					2.77	7.5
TRINITY_DN37680_c0_g1	*JMJ14*	Probable lysine-specific demethylase JMJ14						11.2
TRINITY_DN7266_c0_g2	*LD*	Homeobox protein LUMINIDEPENDENS	2.16		2.68		3.09	7.25
TRINITY_DN6935_c0_g1	*MSI4*	WD-40 repeat-containing protein MSI4						10.21
TRINITY_DN34495_c0_g1	*NFYB2*	Nuclear transcription factor Y subunit B-2			2.4	2.58	2.59	
TRINITY_DN18038_c0_g1	*NFYB3*	Nuclear transcription factor Y subunit B-3					2.17	8.27
TRINITY_DN35701_c0_g2	*NFYC1*	Nuclear transcription factor Y subunit C-1						2.59
TRINITY_DN1267_c0_g1	*PEX4*	Pollen-specific leucine-rich repeat extensin-like protein 4	−8.5	−5.05	−5.26	−8.04	−4.31	−3.62
TRINITY_DN6353_c0_g1	*PHYA*	Phytochrome A			2.15	2.39		
TRINITY_DN21387_c0_g1	*PHYB*	Phytochrome B	−2.28					
TRINITY_DN7954_c0_g1	*PI*	Floral homeotic protein PISTILLATA	−8.44	−6.51	−7.06	−8.3	−3.54	−3.36
TRINITY_DN12744_c0_g2	*SEP1*	Developmental protein SEPALLATA 1	−7.94	−7.09	−7.56	−8.08	−2.8	
TRINITY_DN31040_c0_g1	*SEP3*	Developmental protein SEPALLATA 3					−2.37	
TRINITY_DN2251_c0_g1	*SOC1*	MADS-box protein SOC1	4.05	2.9	3.46	3.1	2.6	12.47
TRINITY_DN6192_c0_g1	*SPL15*	Squamosa promoter-binding-like protein 15						10.17
TRINITY_DN6705_c0_g1	*SPL3*	Squamosa promoter-binding-like protein 3	−3.84		−3.38	−4.54		−2.82
TRINITY_DN14821_c0_g1	*SPL4*	Squamosa promoter-binding-like protein 4		−3.38	−2.18			
TRINITY_DN57302_c0_g1	*SRF6*	Protein STRUBBELIG-RECEPTOR FAMILY 6	−4.07	−3.28	−4.01	−3.91		
TRINITY_DN46547_c0_g1	*SRF8*	Protein STRUBBELIG-RECEPTOR FAMILY 8						7.42
TRINITY_DN2130_c0_g1	*SRR1*	Protein SENSITIVITY TO RED LIGHT REDUCED 1						12.09
TRINITY_DN29363_c0_g1	*SVP*	MADS-box protein SVP						−3.8
TRINITY_DN5406_c1_g1	*UBC1*	Ubiquitin-conjugating enzyme E2 1						7.86
TRINITY_DN16649_c0_g1	*ULT1*	Protein ULTRAPETALA 1					−2.18	
TRINITY_DN13113_c0_g1	*VRN1*	B3 domain-containing transcription factor VRN1	−3.2	−2.14	−2.41	−2.73		
TRINITY_DN8011_c0_g1	*WNK1*	Serine/threonine-protein kinase WNK1						8.81

**Table 5 genes-14-00901-t005:** List of differentially expressed genes (DEGs) in ovules of *Limonium* plants, namely apomictic *L. multiflorum* (M) relative to sexual *L. auriculifolium* (a) and *L. ovalifolium* (o), in different stages, which are involved in tryptophan, ethylene, aminoacyl-tRNA, or lysine metabolism. DEGs were filtered by |log2 fold-change (log2FC)| > 2. DEGs were mapped to their respective *A. thaliana* homolog (gene name) and annotated according to its reference genome (red: up-regulated DEGs; blue: down-regulated DEGs).

Gene ID	Gene Name	Protein Name	Log2FC						
			M1a1	M1a2	M1o1	M1o2	M2a4	M2o4	M2d4
**Aminoacyl-tRNA**									
TRINITY_DN2761_c0_g1	*AO*	L-aspartate oxidase, chloroplastic	2.11		2.61				
TRINITY_DN4987_c0_g1	*EDD1*	Glycine--tRNA ligase, chloroplastic/mitochondrial 2						12.19	
TRINITY_DN157005_c0_g1	*GDH2*	Glycine cleavage system H protein 2, mitochondrial	−3.49	−2.94	−2.46	−2.49	−2.5		
TRINITY_DN2155_c4_g1	*GLDP1*	Glycine dehydrogenase (decarboxylating) 1, mitochondrial						2.29	
TRINITY_DN10603_c0_g1	*GLDP2*	Glycine dehydrogenase (decarboxylating) 2, mitochondrial		4.14				7.09	
TRINITY_DN2985_c0_g1	*GRDP2*	Glycine-rich domain-containing protein 2						−2.18	
TRINITY_DN6537_c0_g1	*PSS1*	CDP-diacylglycerol--serine O-phosphatidyltransferase 1					9.39		
TRINITY_DN20064_c0_g1	*RBG3*	Glycine-rich RNA-binding protein 3, mitochondrial						−2.44	
TRINITY_DN8570_c0_g1	*RBG4*	Glycine-rich RNA-binding protein 4, mitochondrial					3.75	8.48	
TRINITY_DN8373_c0_g1	*RBG5*	Glycine-rich RNA-binding protein 5, mitochondrial						9.41	
TRINITY_DN27069_c0_g1	*RZ1A*	Glycine-rich RNA-binding protein RZ1A						2.24	
TRINITY_DN14144_c0_g1	*UGLYAH*	(S)-ureidoglycine aminohydrolase	2.79			2.01	3.21	7.94	
**Ethylene**									
TRINITY_DN15258_c0_g1	*AIL5*	AP2-like ethylene-responsive transcription factor AIL5	−2.78	−3.27	−3.23	−2.23		8.8	
TRINITY_DN51613_c0_g1	*ANT*	AP2-like ethylene-responsive transcription factor ANT						6.87	
TRINITY_DN13736_c0_g1	*CRF2*	Ethylene-responsive transcription factor CRF2					−3.02	−2.65	
TRINITY_DN68_c0_g2	*EIN2*	Ethylene-insensitive protein 2		2.15	2.3			11.11	
TRINITY_DN2468_c0_g1	*ERF010*	Ethylene-responsive transcription factor ERF010						3.25	
TRINITY_DN9993_c0_g1	*ERF014*	Ethylene-responsive transcription factor ERF014						−2.01	
TRINITY_DN1685_c1_g1	*ERF018*	Ethylene-responsive transcription factor ERF018			2.98	3.3	3.09	2.52	
TRINITY_DN18951_c0_g1	*ERF034*	Ethylene-responsive transcription factor ERF034						−3.47	
TRINITY_DN7978_c0_g2	*ERF054*	Ethylene-responsive transcription factor ERF054	3.29		3.21	4.29	3.91	10.81	
TRINITY_DN455_c0_g2	*ERF061*	Ethylene-responsive transcription factor ERF061			2.36			10.86	
TRINITY_DN74944_c0_g3	*ERF071*	Ethylene-responsive transcription factor ERF071						−2.59	
TRINITY_DN2504_c1_g1	*ERF109*	Ethylene-responsive transcription factor ERF109	2.63			3.22	4.14	10.25	
TRINITY_DN12655_c0_g1	*ERF114*	Ethylene-responsive transcription factor ERF114						9.96	
TRINITY_DN40207_c0_g1	*ERF118*	Ethylene-responsive transcription factor ERF118					−2.01		
TRINITY_DN15135_c0_g1	*ERF1A*	Ethylene-responsive transcription factor 1A						11.43	
TRINITY_DN5700_c2_g1	*ERF2*	Ethylene-responsive transcription factor 2					2.19	10.77	
TRINITY_DN10435_c0_g1	*ERF5*	Ethylene-responsive transcription factor 5		7.93	7.95			7.89	
TRINITY_DN4039_c0_g2	*ERF9*	Ethylene-responsive transcription factor 9						2.46	
TRINITY_DN9095_c0_g1	*ERS1*	Ethylene response sensor 1						9.71	
TRINITY_DN2665_c0_g1	*RAP2-13*	Ethylene-responsive transcription factor RAP2-13						10.11	
TRINITY_DN40774_c0_g3	*RAP2-3*	Ethylene-responsive transcription factor RAP2-3	−5.92	−4.04	−5.69	−5.82		−2.5	
TRINITY_DN1497_c0_g1	*RAP2-4*	Ethylene-responsive transcription factor RAP2-4						4.08	
TRINITY_DN2_c1_g1	*RAP2-6*	Ethylene-responsive transcription factor RAP2-6			2.25	2.3	2.41	11.76	
TRINITY_DN12350_c0_g1	*RAP2-7*	Ethylene-responsive transcription factor RAP2-7	6.43	4.91	6.54	4.61	4.12	9.24	
TRINITY_DN5947_c0_g1	*RTE1*	Protein REVERSION-TO-ETHYLENE SENSITIVITY1						10.91	
TRINITY_DN29694_c0_g1	*SHN3*	Ethylene-responsive transcription factor SHINE 3					−3.72	−3.93	
TRINITY_DN4330_c3_g1	*TINY*	Ethylene-responsive transcription factor TINY						9.64	
TRINITY_DN148_c3_g1	*WRI1*	Ethylene-responsive transcription factor WRI1	−2.13			−2.63			
TRINITY_DN5382_c0_g1	*AT4G13040*	Ethylene-responsive transcription factor-like protein At4g13040						9.5	
**Lysine**									
TRINITY_DN6270_c0_g1	*AATL1*	Lysine histidine transporter-like 8						2.04	
TRINITY_DN8133_c0_g1	*ASHR1*	Histone-lysine N-methyltransferase ASHR1						10.07	
TRINITY_DN4218_c0_g3	*ATX2*	Histone-lysine N-methyltransferase ATX2						7.65	−3.53
TRINITY_DN3010_c0_g1	*ATX4*	Histone-lysine N-methyltransferase ATX4						11.71	
TRINITY_DN209_c0_g2	*ATX5*	Histone-lysine N-methyltransferase ATX5					7.64	7.46	
TRINITY_DN1076_c0_g1	*ATXR2*	Histone-lysine N-methyltransferase ATXR2						10.78	
TRINITY_DN1656_c0_g1	*ATXR3*	Histone-lysine N-methyltransferase ATXR3						2.13	
TRINITY_DN11945_c0_g1	*ATXR4*	Histone-lysine N-methyltransferase ATXR4						9.06	
TRINITY_DN15991_c0_g1	*ATXR5*	Histone-lysine N-methyltransferase ATXR5					−3.1	6.81	
TRINITY_DN22923_c0_g1	*ELF6*	Probable lysine-specific demethylase ELF6						5.1	
TRINITY_DN6877_c1_g1	*EMB3003*	Dihydrolipoyllysine-residue acetyltransferase component 5 of pyruvate dehydrogenase complex. chloroplastic						−2.46	
TRINITY_DN14929_c0_g1	*EZA1*	Histone-lysine N-methyltransferase EZA1						2.92	
TRINITY_DN37680_c0_g1	*JMJ14*	Probable lysine-specific demethylase JMJ14						11.2	
TRINITY_DN3464_c0_g3	*JMJ25*	Lysine-specific demethylase JMJ25						8.39	
TRINITY_DN3267_c0_g1	*JMJ30*	Lysine-specific demethylase JMJ30			−2.4	−2.16		9.65	
TRINITY_DN1796_c2_g1	*LHT1*	Lysine histidine transporter 1	4.61	3.22	2.51	3.81	4.38	10.07	
TRINITY_DN1796_c0_g3	*LHT2*	Lysine histidine transporter 2						9.7	
TRINITY_DN6877_c1_g2	*LTA2*	Dihydrolipoyllysine-residue acetyltransferase component 4 of pyruvate dehydrogenase complex. chloroplastic	−2.23						
TRINITY_DN92_c0_g1	*OVA5*	Lysine--tRNA ligase. chloroplastic/mitochondrial						2.08	
TRINITY_DN22984_c0_g1	*SUVH6*	Histone-lysine N-methyltransferase. H3 lysine-9 specific SUVH6					−2.35	7.17	
TRINITY_DN393_c0_g2	*SUVH9*	Histone-lysine N-methyltransferase family member SUVH9						10.36	
TRINITY_DN18492_c0_g1	*SUVR1*	Probable inactive histone-lysine N-methyltransferase SUVR1						10.33	
TRINITY_DN10785_c0_g1	*SUVR3*	Histone-lysine N-methyltransferase SUVR3						7.14	
TRINITY_DN27123_c0_g1	*SUVR4*	Histone-lysine N-methyltransferase SUVR4						9.46	
TRINITY_DN22651_c0_g1	*AT1G25530*	Lysine histidine transporter-like 6						10.9	2.27
TRINITY_DN21036_c0_g1	*AT4G26910*	Dihydrolipoyllysine-residue succinyltransferase component of 2-oxoglutarate dehydrogenase complex 2. mitochondrial						9.63	
TRINITY_DN1947_c0_g1	*AT5G55070*	Dihydrolipoyllysine-residue succinyltransferase component of 2-oxoglutarate dehydrogenase complex 1. mitochondrial						9.52	
TRINITY_DN971_c1_g3	*AT4G35180*	Lysine histidine transporter-like 7						8.3	
TRINITY_DN40168_c0_g1	*AT3G11710*	Lysine--tRNA ligase. cytoplasmic						7.18	
**Tryptophan**									
TRINITY_DN10705_c0_g1	*TAA1*	L-tryptophan--pyruvate aminotransferase 1						7.41	
TRINITY_DN6324_c0_g2	*TAR2*	Tryptophan aminotransferase-related protein 2	−4.07	−2.65	−3.72	−4.13	−3.64		
TRINITY_DN18489_c0_g3	*TSB2*	Tryptophan synthase β chain 2. chloroplastic					2.13	7.83	
TRINITY_DN257_c0_g2	*AT3G04600*	Tryptophan--tRNA ligase. cytoplasmic						10.25	

**Table 6 genes-14-00901-t006:** Changes in expression of uniquely annotated differentially expressed genes (DEGs) related to pollen from ovules of apomictic *L. multiflorum* (M) in S1 (1) relative to sexual *L. auriculifolium* (a) and *L. ovalifolium* (o) in either S1 (1) or S2 (2), and *L. multiflorum* (M) in S2 (2) relative to *L. auriculifolium* (a), *L. ovalifolium* (o), and facultative apomictic *L. dodartii* (d) in S3/S4 (4). Annotated DEGs according to *A. thaliana* homologs (gene names) were searched in biological process (BP), molecular function (MF), and cellular component (CC) Gene Ontology (GO) terms related to pollen (red: up-regulated DEGs; blue: down-regulated DEGs).

Gene ID	Gene Name	Protein Name	M1a1	M1a2	M1o1	M1o2	M2a4	M2o4	M2d4
**pollen development (GO:0009555)**									
TRINITY_DN5448_c0_g1	*NAS3*	Nicotianamine synthase 3	−3.49	−4.95	−3.84	−3.52			
TRINITY_DN95402_c0_g1	*PS1*	FHA domain-containing protein PS1	−2.61			−2.08	−2.42	8.49	
TRINITY_DN38119_c0_g1	*KDSB*	3-deoxy-manno-octulosonate cytidylyltransferase, mitochondrial	2.07						
TRINITY_DN10816_c0_g1	*TMK3*	Receptor-like kinase TMK3	−3.51			−2.75	−2.57	−3.57	
TRINITY_DN923_c0_g1	*CALS5*	Callose synthase 5	−2.91		−2.05	−3.3	−3.49		
TRINITY_DN111580_c0_g2	*CYP73A5*	Trans-cinnamate 4-monooxygenase	2.6	2.57	3.63	2.01			
TRINITY_DN41226_c0_g1	*PAL1*	Phenylalanine ammonia-lyase 1	−2.27			−2.65		−3.8	
TRINITY_DN2633_c0_g1	*CALS9*	Callose synthase 9	2.08				2.07	9.17	
TRINITY_DN1417_c2_g1	*FAB1B*	1-phosphatidylinositol-3-phosphate 5-kinase FAB1B	2.35		2.38	2.68		3.2	
TRINITY_DN1233_c4_g1	*AT4G39110*	Probable receptor-like protein kinase At4g39110	−8.01	−4.86	−4.65	−7.56	−4.44	−3.74	
TRINITY_DN3115_c1_g1	*ATL73*	RING-H2 finger protein ATL73	−2.87	−2.93	−2.79	−2.45		8.09	
TRINITY_DN6027_c0_g1	*LRP1*	Protein LATERAL ROOT PRIMORDIUM 1	−2.64	−2.92	−2.65	−2.47			
TRINITY_DN5863_c0_g1	*LCB2A*	Long chain base biosynthesis protein 2a	2.81						
TRINITY_DN7884_c0_g1	*XRI1*	Protein XRI1	−4.22			−3.9	−2.9		
TRINITY_DN4445_c0_g1	*CEP1*	KDEL-tailed cysteine endopeptidase CEP1	−6.27	−7.42	−6.31	−5.23			
TRINITY_DN8949_c0_g1	*SWEET13*	Bidirectional sugar transporter SWEET13	−2.61	−2.07		−2.58			
TRINITY_DN10107_c0_g1	*ABCB25*	ABC transporter B family member 25, mitochondrial	2.48		2.73		2.35	8.62	
TRINITY_DN27954_c0_g3	*LECRK42*	L-type lectin-domain containing receptor kinase IV.2		3.43				−2.54	−3.02
TRINITY_DN8056_c1_g1	*CALS11*	Callose synthase 11		3.3	3.44		3.29	8.01	
TRINITY_DN4094_c0_g1	*NMT1*	Phosphoethanolamine N-methyltransferase 1		2.15	2.43				
TRINITY_DN17932_c0_g1	*WRKY2*	Probable WRKY transcription factor 2			−2.02	−2.22	−2.23	7.15	
TRINITY_DN12987_c0_g1	*PIN5*	Auxin efflux carrier component 5			2.41			10	3.48
TRINITY_DN4445_c0_g1	*CEP1*	KDEL-tailed cysteine endopeptidase CEP1							
TRINITY_DN5448_c0_g1	*NAS3*	Nicotianamine synthase 3							
TRINITY_DN3873_c0_g1	*AGL65*	Agamous-like MADS-box protein AGL65				−2.29	−2.62	−3.28	
TRINITY_DN47156_c0_g1	*IPK2B*	Inositol polyphosphate multikinase β				3.28		7.44	
TRINITY_DN2407_c1_g1	*BZIP34*	Basic leucine zipper 34					−5.18	−2.68	
TRINITY_DN12180_c0_g1	*AGL66*	Agamous-like MADS-box protein AGL66					−4.72	−3.81	
TRINITY_DN15991_c0_g1	*ATXR5*	Histone-lysine N-methyltransferase ATXR5					−3.1	6.81	
TRINITY_DN9971_c0_g1	*MYB80*	Transcription factor MYB80					4.46	8.63	
TRINITY_DN10584_c0_g1	*MCM7*	DNA replication licensing factor MCM7					−2.04		
TRINITY_DN5641_c0_g1	*LOX3*	Lipoxygenase 3, chloroplastic					2.93	2.75	
TRINITY_DN23341_c0_g1	*D6PKL3*	Serine/threonine-protein kinase D6PKL3					2.16	8.44	
TRINITY_DN11273_c0_g1	*MCM4*	DNA replication licensing factor MCM4						−4.4	
TRINITY_DN440_c0_g1	*MCM8*	Probable DNA helicase MCM8						12.61	
TRINITY_DN3132_c0_g1	*PGDH1*	D-3-phosphoglycerate dehydrogenase 1, chloroplastic						11.73	
TRINITY_DN12133_c0_g1	*MRS2-2*	Magnesium transporter MRS2-2						11.26	
TRINITY_DN7153_c0_g1	*APD2*	E3 ubiquitin-protein ligase APD2						11.16	
TRINITY_DN4731_c0_g3	*MYB101*	Transcription factor MYB101						11.05	
TRINITY_DN1371_c0_g1	*RGTB1*	Geranylgeranyl transferase type-2 subunit β 1						11.03	
TRINITY_DN2976_c0_g1	*AT2G21870*	Probable ATP synthase 24 kDa subunit. mitochondrial						9.59	
TRINITY_DN11362_c1_g1	*WRKY35*	Probable WRKY transcription factor 35						9.53	
TRINITY_DN166676_c0_g1	*CER26L*	Protein ECERIFERUM 26-like						9.43	
TRINITY_DN9251_c0_g1	*FAS1*	Chromatin assembly factor 1 subunit FAS1						9.15	
TRINITY_DN17928_c0_g2	*CYP94B3*	Cytochrome P450 94B3						−2.3	
TRINITY_DN1593_c0_g1	*DSE1*	Protein DECREASED SIZE EXCLUSION LIMIT 1						3.87	
TRINITY_DN41732_c1_g1	*GAPCP1*	Glyceraldehyde-3-phosphate dehydrogenase GAPCP1, chloroplastic						−2.38	
TRINITY_DN968_c1_g1	*CDKA-1*	Cyclin-dependent kinase A-1						8.61	
TRINITY_DN114534_c0_g1	*LCB2B*	Long chain base biosynthesis protein 2b						8.44	
TRINITY_DN39521_c0_g1	*RUK*	Serine/threonine-protein kinase RUNKEL						8.41	
TRINITY_DN30938_c0_g1	*XPO1*	Protein EXPORTIN 1A						2.82	
TRINITY_DN54834_c0_g2	*KIN7A*	Kinesin-like protein KIN-7A						7.81	
TRINITY_DN29594_c0_g1	*PTD*	Protein PARTING DANCERS						7.27	
TRINITY_DN3033_c0_g4	*P5CSA*	Delta-1-pyrroline-5-carboxylate synthase A						2.7	
TRINITY_DN31412_c0_g1	*RGP1*	UDP-arabinopyranose mutase 1						7.16	
TRINITY_DN14808_c0_g1	*RTEL1*	Regulator of telomere elongation helicase 1 homolog						6.99	
TRINITY_DN49207_c0_g1	*SRS1*	Protein SHI RELATED SEQUENCE 1						6.96	
TRINITY_DN29704_c0_g5	*SRS5*	Protein SHI RELATED SEQUENCE 5						6.81	
TRINITY_DN9646_c0_g1	*ATRX*	Protein CHROMATIN REMODELING 20						2.32	
TRINITY_DN21539_c0_g1	*BHLH91*	Transcription factor bHLH91						6.74	
TRINITY_DN826_c0_g2	*TULP7*	Tubby-like F-box protein 7						2.15	
TRINITY_DN4508_c0_g1	*MPK4*	Mitogen-activated protein kinase 4						2.09	
TRINITY_DN161_c0_g2	*NEDD1*	Protein NEDD1						2.22	
TRINITY_DN6156_c0_g1	*PKSA*	Type III polyketide synthase A	−3.61		−6.78	−6.63	−3.39	−2.98	
TRINITY_DN31801_c4_g1	*GPAT1*	Glycerol-3-phosphate acyltransferase 1					−4.13	−4.38	
TRINITY_DN17166_c0_g1	*4CL3*	4-coumarate--CoA ligase 3	−4.44		−2.69	−4.06	−3.78	−2.43	
TRINITY_DN16998_c0_g1	*ZAT2*	Zinc finger protein ZAT2					−4.9	−4.15	
TRINITY_DN15578_c0_g1	*LRL1*	Transcription factor LRL1					−3.39	6.94	
TRINITY_DN4588_c0_g2	*ABCG31*	ABC transporter G family member 31	−2.72	−3.81	−2.13	−2.53	−2.22	10.04	
TRINITY_DN45712_c0_g1	*FAR2*	Fatty acyl-CoA reductase 2, chloroplastic	−3.28		−6.24	−6.93	−4.07	−2.8	
TRINITY_DN9952_c0_g1	*ABCG26*	ABC transporter G family member 26					−3.28		
TRINITY_DN15838_c0_g1	*TIP5-1*	Probable aquaporin TIP5-1				−2.48	−2.82	8.26	
TRINITY_DN6869_c0_g1	*A6*	Probable glucan endo-1,3-β-glucosidase A6	−2.6		−7.57	−7.88	−2.61	−2.95	
TRINITY_DN156774_c0_g1	*ABCG9*	ABC transporter G family member 9					−2.19	−3.86	
TRINITY_DN31379_c0_g1	*TKPR1*	Tetraketide α-pyrone reductase 1					−4.21	−4.43	
TRINITY_DN3196_c3_g1	*EMS1*	Leucine-rich repeat receptor protein kinase EMS1					−2.05	−2.34	
TRINITY_DN176325_c0_g1	*COPT1*	Copper transporter 1	−3.79	−3.86	−3.99	−3.76	−2.12		
TRINITY_DN4485_c0_g1	*CYP704B1*	Cytochrome P450 704B1						−2.52	
**microsporogenesis (GO:0009556)**									
TRINITY_DN5204_c0_g1	*PLC2*	Phosphoinositide phospholipase C 2	3.07		2.44	3.37	2.07	3.53	
TRINITY_DN3196_c3_g1	*EMS1*	Leucine-rich repeat receptor protein kinase EMS1					−2.05	−2.34	
TRINITY_DN6537_c0_g1	*PSS1*	CDP-diacylglycerol--serine O-phosphatidyltransferase 1						9.39	
TRINITY_DN2003_c0_g1	*FH14*	Formin-like protein 14						2.33	
**pollen germination (GO:0009846)**									
TRINITY_DN373_c0_g1	*CSLD1*	Cellulose synthase-like protein D1	−3.31			−4.72	−3.45	−2.22	
TRINITY_DN19056_c0_g1	*IP5P13*	Type I inositol polyphosphate 5-phosphatase 13				−2.38	−3.35	−2.36	
TRINITY_DN4602_c0_g1	*JGB*	Protein JINGUBANG					−4	−4.06	
TRINITY_DN8465_c0_g1	*CSLD4*	Cellulose synthase-like protein D4					−3.89	−3.71	
TRINITY_DN46361_c0_g1	*PTF2*	Plant-specific TFIIB-related protein PTF2						9.36	
TRINITY_DN19056_c0_g3	*IP5P12*	Type I inositol polyphosphate 5-phosphatase 12						6.94	
**pollination (GO:0009856)**									
TRINITY_DN6022_c0_g1	*ARPN*	Basic blue protein					−5.23	−3.65	
**pollen tube growth (GO:0009860)**									
TRINITY_DN8457_c0_g3	*AT1G03010*	BTB/POZ domain-containing protein At1g03010	−4.41	−4.84	−4.35	−3.42		2.01	
TRINITY_DN5849_c0_g1	*XI-E*	Myosin-11	2.14	4.3	3.52			7.48	
TRINITY_DN691_c0_g2	*ARAC5*	Rac-like GTP-binding protein ARAC5	−3.62	−2.02	−2.9	−3.64	−2.65		
TRINITY_DN11455_c0_g1	*CBL1*	Calcineurin B-like protein 1	2.24					4.42	
TRINITY_DN1267_c0_g1	*PEX4*	Pollen-specific leucine-rich repeat extensin-like protein 4	−8.5	−5.05	−5.26	−8.04	−4.31	−3.62	
TRINITY_DN677_c0_g2	*NPF8.2*	Protein NRT1/PTR FAMILY 8,2	−2.24	−2.05	−2.36	−2.43			
TRINITY_DN7934_c0_g1	*ABCG28*	ABC transporter G family member 28	−2.78			−3.68	−4.42	7.31	
TRINITY_DN5923_c0_g1	*OASA1*	Cysteine synthase 1		2.1	2.13			9.71	
TRINITY_DN8399_c0_g1	*RIC5*	CRIB domain-containing protein RIC5					−4.82	−4.6	
TRINITY_DN8450_c0_g1	*PPME1*	Pectinesterase PPME1					−4.77		
TRINITY_DN5260_c2_g1	*PEX1*	Pollen-specific leucine-rich repeat extensin-like protein 1					−4.49	−3.64	
TRINITY_DN16608_c0_g2	*RIC6*	CRIB domain-containing protein RIC6					−4.2	−3.93	
TRINITY_DN184921_c0_g1	*AGC1-5*	Serine/threonine-protein kinase AGC1-5					−4.13		
TRINITY_DN35118_c0_g1	*CNGC18*	Cyclic nucleotide-gated ion channel 18					−3.97	6.14	
TRINITY_DN14636_c1_g1	*CNGC7*	Putative cyclic nucleotide-gated ion channel 7					−3.93	−4.53	
TRINITY_DN15772_c0_g3	*CXE18*	Probable carboxylesterase 18					−3.71		
TRINITY_DN8917_c0_g2	*TOPP8*	Serine/threonine-protein phosphatase PP1 isozyme 8					−3.98		
TRINITY_DN12354_c0_g2	*KLCR2*	Protein KINESIN LIGHT CHAIN-RELATED 2					−3.52	−3.57	
TRINITY_DN43059_c0_g1	*TCTP1*	Translationally-controlled tumor protein 1					−2.39	6.64	
TRINITY_DN145_c0_g1	*AT2G41970*	Probable protein kinase At2g41970					−2.03		
TRINITY_DN8899_c0_g3	*ARAC11*	Rac-like GTP-binding protein ARAC11					−2		
TRINITY_DN3041_c0_g3	*CDI*	Protein CDI						11.87	
TRINITY_DN10453_c0_g1	*AGC1-7*	Serine/threonine-protein kinase AGC1-7						11.84	
TRINITY_DN7093_c0_g1	*PIGA*	Phosphatidylinositol N-acetylglucosaminyltransferase subunit A						10.57	
TRINITY_DN1136_c1_g1	*FIM5*	Fimbrin-5						10.5	
TRINITY_DN17922_c0_g2	*WIP2*	Zinc finger protein WIP2						−2.58	
TRINITY_DN6837_c0_g1	*XI-C*	Myosin-9						9.53	
TRINITY_DN9379_c0_g1	*SBT3.1*	Subtilisin-like protease SBT3.1						−2.11	
TRINITY_DN12065_c0_g1	*MIRO1*	Mitochondrial Rho GTPase 1						8.43	
**pollen tube adhesion (GO:0009865)**									
TRINITY_DN14184_c0_g1	*HS1*	Stress-response A/B barrel domain-containing protein HS1	−3.03			−2.2	−2.58	−2.22	
**pollen-pistil interaction (GO:0009875)**									
TRINITY_DN2016_c0_g1	*MPK3*	Mitogen-activated protein kinase 3	2.1				2.21	4.22	
TRINITY_DN4521_c0_g1	*MAA3*	Probable helicase MAGATAMA 3						12.13	
TRINITY_DN14556_c0_g1	*MKK9*	Mitogen-activated protein kinase kinase 9						9.61	
**pollen maturation (GO:0010152)**									
TRINITY_DN8479_c0_g1	*DRP1C*	Dynamin-related protein 1C						13.13	
TRINITY_DN13085_c0_g1	*AFB2*	Protein AUXIN SIGNALING F-BOX 2						9.73	
TRINITY_DN6009_c0_g1	*RPK2*	LRR receptor-like serine/threonine-protein kinase RPK2						3.81	
TRINITY_DN3540_c1_g1	*PDR2*	Probable manganese-transporting ATPase PDR2						2.12	
**pollen tube guidance (GO:0010183)**									
TRINITY_DN6442_c0_g1	*A39*	Aspartic proteinase 39	−2.79			−3.46	−3.7		
TRINITY_DN253_c1_g1	*COBL10*	COBRA-like protein 10	−8.86	−5.63	−5.6	−8.42	−4.18		
TRINITY_DN22116_c0_g1	*MIK2*	MDIS1-interacting receptor like kinase 2	2.62					8.29	−2.01
TRINITY_DN17193_c0_g1	*MIK1*	MDIS1-interacting receptor like kinase 1	−4.33			−3.54	−3.34	−4.58	
TRINITY_DN253_c0_g2	*COBL11*	COBRA-like protein 11					−4.68	−3.27	
TRINITY_DN30272_c0_g1	*GEX3*	Protein GAMETE EXPRESSED 3					−2.97		
TRINITY_DN20396_c0_g1	*LIP2*	Receptor-like kinase LIP2						9.71	
TRINITY_DN7012_c0_g1	*SIZ1*	E3 SUMO-protein ligase SIZ1						2.21	
TRINITY_DN5591_c0_g1	*POD1*	Protein POLLEN DEFECTIVE IN GUIDANCE 1						2.11	
**pollen wall assembly (GO:0010208)**									
TRINITY_DN6156_c0_g1	*PKSA*	Type III polyketide synthase A	−3.61		−6.78	−6.63	−3.39	−2.98	
TRINITY_DN17166_c0_g1	*4CL3*	4-coumarate--CoA ligase 3	−4.44		−2.69	−4.06	−3.78	−2.43	
TRINITY_DN923_c0_g1	*CALS5*	Callose synthase 5	−2.91		−2.05	−3.3	−3.49		
TRINITY_DN4588_c0_g2	*ABCG31*	ABC transporter G family member 31	−2.72	−3.81	−2.13	−2.53	−2.22	10.04	
TRINITY_DN45712_c0_g1	*FAR2*	Fatty acyl-CoA reductase 2. chloroplastic	−3.28		−6.24	−6.93	−4.07	−2.8	
TRINITY_DN9952_c0_g1	*ABCG26*	ABC transporter G family member 26					−3.28		
TRINITY_DN6869_c0_g1	*A6*	Probable glucan endo-1,3-β-glucosidase A6	−2.6		−7.57	−7.88	−2.61	−2.95	
TRINITY_DN31379_c0_g1	*TKPR1*	Tetraketide α-pyrone reductase 1					−4.21	−4.43	
TRINITY_DN156774_c0_g1	*ABCG9*	ABC transporter G family member 9					−2.19	−3.86	
TRINITY_DN3259_c0_g1	*AHL16*	AT-hook motif nuclear-localized protein 16	2.36	2.17		2.08	2.17		
TRINITY_DN6293_c0_g2	*CYP703A2*	Cytochrome P450 703A2						−3.11	
TRINITY_DN4485_c0_g1	*CYP704B1*	Cytochrome P450 704B1						−2.52	
**pollen tube reception (GO:0010483)**									
TRINITY_DN2319_c0_g2	*FER*	Receptor-like protein kinase FERONIA	−3.33	−2.02		−2.9	−2.85	−3.99	
TRINITY_DN1919_c0_g1	*EVN*	Dolichol kinase EVAN						3.02	
**pollen exine formation (GO:0010584)**									
TRINITY_DN6156_c0_g1	*PKSA*	Type III polyketide synthase A	−3.61		−6.78	−6.63	−3.39	−2.98	
TRINITY_DN17166_c0_g1	*4CL3*	4-coumarate--CoA ligase 3	−4.44		−2.69	−4.06	−3.78	−2.43	
TRINITY_DN45712_c0_g1	*FAR2*	Fatty acyl-CoA reductase 2. chloroplastic	−3.28		−6.24	−6.93	−4.07	−2.8	
TRINITY_DN6869_c0_g1	*A6*	Probable glucan endo-1,3-β-glucosidase A6	−2.6		−7.57	−7.88	−2.61	−2.95	
TRINITY_DN4186_c0_g1	*QRT3*	Polygalacturonase QRT3	−4.02	−3.09	−4.76	−4.77			
TRINITY_DN1609_c0_g1	*BRI1*	Protein BRASSINOSTEROID INSENSITIVE 1	−2.45			−2.2			
TRINITY_DN10341_c0_g1	*PKSB*	Type III polyketide synthase B			−2.2				
TRINITY_DN31379_c0_g1	*TKPR1*	Tetraketide α-pyrone reductase 1					−4.21	−4.43	
TRINITY_DN9952_c0_g1	*ABCG26*	ABC transporter G family member 26					−3.28		
TRINITY_DN6293_c0_g2	*CYP703A2*	Cytochrome P450 703A2						−3.11	
TRINITY_DN17210_c0_g1	*IRX9H*	Probable β-1,4-xylosyltransferase IRX9H						10.62	
TRINITY_DN8740_c0_g1	*SHT*	Spermidine hydroxycinnamoyl transferase						9.57	
TRINITY_DN4485_c0_g1	*CYP704B1*	Cytochrome P450 704B1						−2.52	
TRINITY_DN42106_c0_g1	*SSL13*	Protein STRICTOSIDINE SYNTHASE-LIKE 13						8.48	
TRINITY_DN558_c0_g1	*CDKG1*	Cyclin-dependent kinase G1						3.14	
TRINITY_DN53293_c1_g1	*TKPR2*	Tetraketide α-pyrone reductase 2						6.87	
**pollen sperm cell differentiation (GO:0048235)**									
TRINITY_DN176325_c0_g1	*COPT1*	Copper transporter 1	−3.79	−3.86	−3.99	−3.76	−2.12		
TRINITY_DN16998_c0_g1	*ZAT2*	Zinc finger protein ZAT2					−4.9	−4.15	
TRINITY_DN31801_c4_g1	*GPAT1*	Glycerol-3-phosphate acyltransferase 1					−4.13	−4.38	
TRINITY_DN40616_c0_g1	*QRT2*	Polygalacturonase QRT2						6.55	−2.45
**recognition of pollen (GO:0048544)**									
TRINITY_DN10463_c0_g1	*SD129*	G-type lectin S-receptor-like serine/threonine-protein kinase SD1-29	3.45	3.42					
TRINITY_DN6512_c0_g1	*SD16*	Receptor-like serine/threonine-protein kinase SD1-6	2.13				2.14	2.61	
TRINITY_DN5252_c0_g2	*AT5G24080*	G-type lectin S-receptor-like serine/threonine-protein kinase At5g24080			2.26	−2.9			
TRINITY_DN38630_c0_g1	*SD11*	G-type lectin S-receptor-like serine/threonine-protein kinase SD1-1				−2.08		8.92	
TRINITY_DN2975_c0_g2	*SD18*	Receptor-like serine/threonine-protein kinase SD1-8						−3.11	
TRINITY_DN21172_c0_g1	*RDR6*	RNA-dependent RNA polymerase 6						−2.18	
TRINITY_DN10167_c0_g1	*AT5G03700*	PAN domain-containing protein At5g03700						−2.38	
TRINITY_DN151527_c0_g1	*AT4G27290*	G-type lectin S-receptor-like serine/threonine-protein kinase At4g27290						7.03	
**microsporocyte nucleus (GO:0048556)**									
TRINITY_DN42827_c0_g1	*ARID1*	AT-rich interactive domain-containing protein 1						10.37	
**pollen tube development (GO:0048868)**									
TRINITY_DN10227_c0_g2	*CSLC12*	Probable xyloglucan glycosyltransferase 12	−5.27	−3.31	−4.36	−5.35	−3.3		
TRINITY_DN2830_c0_g1	*EDA30*	Protein EMBRYO SAC DEVELOPMENT ARREST 30						11.59	
TRINITY_DN16425_c0_g1	*LPPG*	Lipid phosphate phosphatase γ						11.3	
TRINITY_DN1186_c1_g2	*KCS5*	3-ketoacyl-CoA synthase 5						−3.15	
TRINITY_DN65496_c0_g1	*E1-β-2*	Pyruvate dehydrogenase E1 component subunit β-3, chloroplastic						9.78	
TRINITY_DN14636_c1_g1	*CNGC7*	Putative cyclic nucleotide-gated ion channel 7					−3.93	−4.53	
TRINITY_DN8450_c0_g1	*PPME1*	Pectinesterase PPME1					−4.77		
TRINITY_DN691_c0_g2	*ARAC5*	Rac-like GTP-binding protein ARAC5	−3.62	−2.02	−2.9	−3.64	−2.65		
TRINITY_DN5487_c0_g1	*ROPGEF12*	Rop guanine nucleotide exchange factor 12					−5.33	−2.79	
TRINITY_DN923_c0_g1	*CALS5*	Callose synthase 5	−2.91		−2.05	−3.3	−3.49		
TRINITY_DN16608_c0_g2	*RIC6*	CRIB domain-containing protein RIC6					−4.2	−3.93	
TRINITY_DN145_c0_g1	*AT2G41970*	Probable protein kinase At2g41970					−2.03		
TRINITY_DN184921_c0_g1	*AGC1-5*	Serine/threonine-protein kinase AGC1-5					−4.13		
TRINITY_DN43059_c0_g1	*TCTP1*	Translationally-controlled tumor protein 1					−2.39	6.64	
TRINITY_DN1241_c0_g1	*MGP4*	UDP-D-xylose:L-fucose α-1,3-D-xylosyltransferase MGP4						2.88	
TRINITY_DN14184_c0_g1	*HS1*	Stress-response A/B barrel domain-containing protein HS1	−3.03			−2.2	−2.58	−2.22	
TRINITY_DN5260_c2_g1	*PEX1*	Pollen-specific leucine-rich repeat extensin-like protein 1					−4.49	−3.64	
TRINITY_DN8399_c0_g1	*RIC5*	CRIB domain-containing protein RIC5					−4.82	−4.6	
TRINITY_DN12354_c0_g2	*KLCR2*	Protein KINESIN LIGHT CHAIN-RELATED 2					−3.52	−3.57	
TRINITY_DN8899_c0_g3	*ARAC11*	Rac-like GTP-binding protein ARAC11					−2		
TRINITY_DN17193_c0_g1	*MIK1*	MDIS1-interacting receptor like kinase 1	−4.33			−3.54	−3.34	−4.58	
TRINITY_DN1267_c0_g1	*PEX4*	Pollen-specific leucine-rich repeat extensin-like protein 4	−8.5	−5.05	−5.26	−8.04	−4.31	−3.62	
TRINITY_DN35118_c0_g1	*CNGC18*	Cyclic nucleotide-gated ion channel 18					−3.97	6.14	
TRINITY_DN30272_c0_g1	*GEX3*	Protein GAMETE EXPRESSED 3					−2.97		
TRINITY_DN15772_c0_g3	*CXE18*	Probable carboxylesterase 18					−3.71		
TRINITY_DN8917_c0_g2	*TOPP8*	Serine/threonine-protein phosphatase PP1 isozyme 8					−3.98		
**microgametogenesis (GO:0055046)**									
TRINITY_DN3449_c0_g1	*PIRL3*	Plant intracellular Ras-group-related LRR protein 3	2.7		2.74		3.21	2.39	
TRINITY_DN15578_c0_g1	*LRL1*	Transcription factor LRL1					−3.39	6.94	
TRINITY_DN4115_c0_g1	*MYB35*	Transcription factor MYB35						10.32	
TRINITY_DN593_c1_g2	*KIN12A*	Kinesin-like protein KIN-12A						8.1	
TRINITY_DN7491_c0_g1	*AUG6*	AUGMIN subunit 6						2.54	
**rejection of self pollen (GO:0060320)**									
TRINITY_DN124729_c0_g1	*SPH5*	S-protein homolog 5					−3.04		
**acceptance of pollen (GO:0060321)**									
TRINITY_DN12342_c0_g1	*SEC5A*	Exocyst complex component SEC5A						2.66	
**pollen coat (GO:0070505)**									
TRINITY_DN4588_c0_g2	*ABCG31*	ABC transporter G family member 31	−2.72	−3.81	−2.13	−2.53	−2.22	10.04	
TRINITY_DN156774_c0_g1	*ABCG9*	ABC transporter G family member 9					−2.19	−3.86	
**regulation of pollen tube growth (GO:0080092)**									
TRINITY_DN28088_c0_g2	*RALFL19*	Protein RALF-like 19	−7.67	−4.99	−4.97	−7.43	−4.92	−4.37	
TRINITY_DN6629_c0_g1	*PRK4*	Pollen receptor-like kinase 4	−6.87	−5.12	−4.56	−6.56	−4.64	−3.32	
TRINITY_DN5487_c0_g1	*ROPGEF12*	Rop guanine nucleotide exchange factor 12					−5.33	−2.79	
TRINITY_DN151480_c0_g1	*CPK24*	Calcium-dependent protein kinase 24					−4.88	−2.89	
TRINITY_DN8879_c0_g1	*CPK17*	Calcium-dependent protein kinase 17					−4.85	−2.86	
TRINITY_DN14172_c0_g1	*PRK3*	Pollen receptor-like kinase 3					−3.48	6.87	
TRINITY_DN9330_c2_g1	*ROPGEF9*	Rop guanine nucleotide exchange factor 9					−3.05		
TRINITY_DN6638_c1_g2	*ROPGEF14*	Rop guanine nucleotide exchange factor 14						−2.1	
TRINITY_DN37662_c0_g1	*RABA4D*	Ras-related protein RABA4d						7.85	
TRINITY_DN4047_c0_g1	*AT1G60420*	Probable nucleoredoxin 1						2.01	
TRINITY_DN3873_c0_g1	*AGL65*	Agamous-like MADS-box protein AGL65				−2.29	−2.62	−3.28	
TRINITY_DN12180_c0_g1	*AGL66*	Agamous-like MADS-box protein AGL66					−4.72	−3.81	
TRINITY_DN923_c0_g1	*CALS5*	Callose synthase 5	−2.91		−2.05	−3.3	−3.49		
TRINITY_DN1233_c4_g1	*AT4G39110*	Probable receptor-like protein kinase At4g39110	−8.01	−4.86	−4.65	−7.56	−4.44	−3.74	
**pollen tube tip (GO:0090404)**									
TRINITY_DN14105_c0_g1	*ALA3*	Phospholipid-transporting ATPase 3	2.51				3.12	10.21	
TRINITY_DN5561_c0_g1	*ANX2*	Receptor-like protein kinase ANXUR2	−7.96	−4.9	−5.05	−7.4	−4.33	−3.5	
**pollen tube (GO:0090406)**									
TRINITY_DN41664_c0_g1	*ZAR1*	Receptor protein kinase-like protein ZAR1	−2.54	−2.17	−2.01	−2.43			
TRINITY_DN6369_c0_g1	*PLT1*	Putative polyol transporter 1	−2.03		−2.04				
TRINITY_DN29983_c0_g1	*AT4G36180*	Probable LRR receptor-like serine/threonine-protein kinase At4g36180	−3.9		−2.83	−4.51			
TRINITY_DN6280_c0_g1	*STP8*	Sugar transport protein 8	−4.78	−2.96	−4.96	−4.45		10.23	
TRINITY_DN4295_c0_g1	*INT2*	Probable inositol transporter 2			2.38			2.79	
TRINITY_DN15838_c0_g1	*TIP5-1*	Probable aquaporin TIP5-1				−2.48	−2.82	8.26	
TRINITY_DN12736_c0_g1	*LLG2*	GPI-anchored protein LLG2					−4.35	−2.64	
TRINITY_DN165958_c0_g1	*GATL4*	Probable galacturonosyltransferase-like 4					−4.18	−4.06	
TRINITY_DN11246_c1_g1	*GNL2*	ARF guanine-nucleotide exchange factor GNL2					−2.8	−2.87	
TRINITY_DN63597_c0_g1	*MDIS2*	Protein MALE DISCOVERER 2					−2.26	9.33	
TRINITY_DN9084_c0_g1	*SUC3*	Sucrose transport protein SUC3						11.54	
TRINITY_DN18602_c0_g1	*SAUR62*	Auxin-responsive protein SAUR62						9.11	
TRINITY_DN38341_c0_g1	*STP7*	Sugar transport protein 7						2.55	

## Data Availability

Raw data are deposited in the NCBI Sequence Read Archive (SRA), under BioProject accession number PRJNA752506. Pipelines and tailored scripts created during this project were made publicly available at https://github.com/ziisabel/CoBiG2/tree/cobig2 (accessed on 1 March 2023).

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
