# Peer review of "Ovule Transcriptome Analysis Discloses Deregulation of Genes and Pathways in Sexual and Apomictic Limonium Species (Plumbaginaceae)"

_genes, 2023, doi:10.3390/genes14040901_

Round 1

Reviewer 1 Report

The work entitled “Ovule transcriptome analysis discloses deregulation of genes and pathways in sexual and apomictic Limonium species (Plumbaginaceae)” proposed to characterize the levels of transcription in the reproductive systems of Limonium, identifying the genes that are differentially expressed in ovules during the production of sexual seeds and apomixis.

The study presents good results and a high impact.

Also, there is good scientific writing.

However, the authors could have validated genes of interest from the ontogenic process of the ovule, increasing the strength of the manuscript; but I understand that the main focus has gone into differential expression.

Authors must remove topic 0 and its respective paragraph (lines 40-46).

Author Response

We thank the Reviewer for his/her comments. We agree that various of the genes identified (Tables) in our study can be potentially used as reference genes to be future validated in quantitative gene expression studies using different development stages of specific tissue types or different reproductive modes. These studies are currently being prepared.

Reviewer 2 Report

Reviewer # Ovule transcriptome analysis discloses deregulation of genes and pathways in sexual and apomictic Limonium species (Plumbaginaceae). The article is good and authors have discussed it fairly with previous findings. It requires minor revision

It is suggested to italicize the scientific names used in the abstract as well.

It is suggested to mention abbreviation at first use of the word. For eg. for APETALA and PISTILLATA use abbreviation in section 2.7 (line 238) instead of mentioning it in discussion section (line 696 and 697). Use the abbreviation consistently throughout the manuscript.

Author Response

We italicize the scientific names used in the abstract.

We abbreviated several genes not only those indicated by the Reviewer and did a general revision of the manuscript. We thank thre reviewer for his/her suggestions